# Dynamic BH3 profiling identifies pro-apoptotic drug combinations for the treatment of malignant pleural mesothelioma

Danielle S. Potter[1,2], Ruochen Du[1,2], Stephan R. Bohl[1,2], Kin-Hoe Chow[3,4], Keith L. Ligon ®[2,3,4,5,6], Raphael Bueno[2,7] & Anthony Letai ®[1,2] ✉

Malignant pleural mesothelioma (MPM) has relatively ineffective first/second-line therapy for advanced disease and only 18% five-year survival for early disease. Drug-induced mitochondrial priming measured by dynamic BH3 profiling identifies efficacious drugs in multiple disease settings. We use high throughput dynamic BH3 profiling (HTDBP) to identify drug combinations that prime primary MPM cells derived from patient tumors, which also prime patient derived xenograft (PDX) models. A navitoclax (BCL-xL/BCL-2/BCL-w antagonist) and AZD8055 (mTORC1/2 inhibitor) combination demonstrates efficacy in vivo in an MPM PDX model, validating HTDBP as an approach to identify efficacious drug combinations. Mechanistic investigation reveals AZD8055 treatment decreases MCL-1 protein levels, increases BIM protein levels, and increases MPM mitochondrial dependence on BCL-xL, which is exploited by navitoclax. Navitoclax treatment increases dependency on MCL-1 and increases BIM protein levels. These findings demonstrate that HTDBP can be used as a functional precision medicine tool to rationally construct combination drug regimens in MPM and other cancers.

Malignant mesothelioma is a rare aggressive neoplasm that is usually associated with prior asbestos exposure and has a generally poor prognosis[1]. The most common type of malignant mesothelioma is pleural, which accounts for 75–80% of cases, followed by peritoneal (20–25%) and tunica vaginalis (<1%), or pericardial (<1%)[2].

There are three main histological subtypes of malignant pleural mesothelioma (MPM): epithelioid, sarcomatoid and biphasic, which is a combination of both epithelioid and sarcomatoid histology. Patients diagnosed with pure epithelioid histology have somewhat better prognosis compared to those with sarcomatoid or biphasic histologies[3]. However, few MPM patients are cured[4,5] and prognosis is poor with an overall survival of 9–17 months after diagnosis[6–8].

Current first-line therapy is multimodal and usually involves surgery for early stage limited disease followed by adjuvant chemotherapy

and/or radiotherapy. Most patients have advanced disease at presentation and receive first-line chemotherapy regimen that include pemetrexed and cisplatin[9]. More recently, the FDA has approved nivolumab in combination with ipilimumab as first-line therapy for unresectable MPM patients. This immunotherapy combination improved median survival to 18.1 months, while patients who underwent chemotherapy survived a median of 14.1 months[10]. 5-year survival of patients undergoing pleurectomy decortication with macroscopic complete resection in a multi-modality setting is approaching 25%[11]. While this progress is encouraging, overall patient survival is mostly dismal and only a small percentage with early disease survive longer than 5 years[12]. Malignant pleural mesothelioma has not benefitted from rationally designed trials based on preclinical data. This is in part due to several factors including a lack of patient derived preclinical models of

[1]Department of Medical Oncology, Dana-Farber Cancer Institute, Boston, MA 02215, USA. [2]Harvard Medical School, Boston, MA 02215, USA. [3]Department of Oncologic Pathology, Dana-Farber Cancer Institute, Boston, MA 02215, USA. [4]Center for Patient Derived Models, Dana-Farber Cancer Institute, Boston, MA 02215, USA. [5]Department of Pathology, Brigham and Women's Hospital, Boston, MA 02215, USA. [6]Cancer Biology Program, Broad Institute of MIT and Harvard, Cambridge, MA 02142, USA. [7]Department of Surgery, Brigham and Women's Hospital, Boston, MA 02115, USA. ✉e-mail: Anthony_Letai@dfci.harvard.edu

MPM and lack of data using modern technological tools to decipher the pharmacodynamic and predictive value of such models.

Mitochondria have a major role in the intrinsic apoptosis pathway, a form of programmed cell death[13]. Activation of intrinsic apoptosis leads to activation of caspases ultimately resulting in cell wide proteolysis, oligonucleosomal DNA cleavage, and cell surface tagging to accelerate phagocytosis[14–16]. The Bcl-2 family of proteins regulates mitochondrial outer member permeabilization (MOMP) and includes anti-apoptotic family members (BCL-2, BCL-xL, BCL-w, MCL-1, BFL-1) and pro-apoptotic family members (effector proteins BAK, BAX and BOK; BH3-only proteins BIM, BID, PUMA, BAD, BMF, NOXA and HRK)[17,18]. The intrinsic apoptotic pathway is activated by MOMP which is a switch-like event, causing the release of apoptotic factors such as cytochrome c which interacts with APAF1 (apoptotic protease activating factor-1), the initiator procaspase 9 and ATP to form a holoenzyme known as the apoptosome. The apoptosome activates caspase 9 which cleaves and activates caspase 3 causing a cascade of caspase activation[16].

Mitochondrial apoptotic priming is a measure of how close to the apoptotic threshold a cell is. A highly primed cell has relatively less anti-apoptotic binding site availability and is closer to the apoptotic threshold than a poorly primed cell, which has more ant-apoptotic availability to buffer an apoptotic assault and is further from the apoptotic threshold[19]. Priming is also regulated by the Bcl-2 family of proteins. The higher the anti-apoptotic to pro-apoptotic ratio, the more availability of anti-apoptotics to neutralize potential pro-apoptotic protein signals efficiently, and the less primed for apoptosis the cell is[19,20]. BH3 profiling is a functional tool that measures mitochondrial apoptotic priming. It uses BH3 peptides derived from the BH3 domain of pro-apoptotic BH3-only proteins to provoke a response from viable mitochondria. Cytochrome c released from the mitochondria after a short incubation with BH3 peptide is used as a surrogate for priming[19]. In general, the more sensitive a mitochondrion is to a BH3 peptide, the more primed it is.

Drugs can alter priming. Dynamic BH3 Profiling (DBP) measures drug-induced changes in priming. A drug treatment that enhances priming will cause mitochondria to undergo MOMP more easily when incubated with a fixed concentration of a promiscuously binding BH3 peptide, such as BIM BH3 peptide, compared to control-treated cells[19–23].

In this work we aim to combine MPM patient samples, MPM patient derived xenograft (PDX) models and DBP to identify drug combinations that prime MPM cells and validate DBP as an approach to identify efficacious drug combinations.

## Results

### Measuring drug-induced apoptotic priming on fresh MPM patient samples

Targeted agents that evoke an early death signal measured by DBP have been shown to be efficacious in vivo[20–23]. We therefore hypothesized that by identifying drug combinations that prime MPM cells ex vivo, we could identify drug/s that would be efficacious in vivo. The schematic of our approach is shown in Fig. 1a. While HTDBP has the capacity to test thousands of drugs simultaneously, given the potentially limited patient tissue in this context, we focused on drugs (and their combinations) clinically relevant (currently in clinical trials or approved in the clinic) to thoracic malignancies in the interest of most efficient clinical translation (Supplementary Table 1).

Prolonged ex vivo culture has been shown to change tumor cell characteristics such as drug sensitivity and gene expression[22,24,25]. A major advantage of DBP is that it requires primary cells to be cultured for less than 24 h, because it measures early changes in apoptotic signaling which occur before frank apoptosis[20–23]. We first set out to see if these standard tissue culture condition had any effect on cell viability. Over 24 h luminescence increased indicating increase in ATP production (Supplementary Fig. 1A).

To identify drug combinations that prime primary MPM cells ex vivo, cells are treated in our CROCS (clinically relevant oncology combination screen), and immunofluorescence microscopy based HTDBP is carried out to identify hits (Fig. 1). For HTDBP to be carried out on limited cells, we first had to determine the optimum BIM BH3 peptide concentration to use in the assay for each sample, chosen to be the $EC_{10}$ for MOMP (10% cytochrome c release) in untreated cells. At this concentration, drug-induced priming can be sensitively captured. We received 13 freshly resected MPM patient tumors, which we dissociated to produce a single cell suspension and seeded on 384-well plate/s. The next day, a BIM BH3 peptide titration on untreated cells to calculate BIM $EC_{10}$ was performed (Supplementary Fig. 1B), followed by HTDBP on CROCS-treated primary MPM cells.

Drug-potentiated peptide-induced loss of cytochrome c was quantified as the difference in percentage of cytochrome c positive cells between DMSO-treated and drug-treated wells; we call this delta priming %. Each treatment (single agent and combination) is present as an experimental duplicate. To test repeatability, we tested the correlation between experimental duplicates (delta priming % n1 Vs. delta priming % n2). All MPM patient samples showed significant correlation between delta priming % replicates (Supplementary Fig. 2A), even when the number of drugs used was low due to low number of cells (mesothelioma patient sample (MPS):D, E, F and G; Supplementary Table 2). Exact drugs used in CROCS for each patient sample is shown in Supplementary Table 3.

We labeled a drug or drug combination as a "hit" in our assay when the treatment scored a mean (of duplicates) Z-score ≥ 3, with neither replicate having a Z-score below 1.5 (red dot Fig. 1b and Supplementary Fig. 2A; gray dashed line marks Z-score = 1.5). All drug combinations and single agents for all MPM patient samples are shown in a heatmap in Supplementary Figure 3, blue is a hit and yellow is a non-hit. This reveals there are clearly some common hits between the patient samples. To assess if chemical vulnerabilities correlated with disease subtype in MPM patient samples, we compared epithelioid and biphasic top hits (only 1 sarcomatoid sample, so no comparisons were made). There was no difference in top hits between epithelioid and biphasic histologies which may be due to the biphasic containing epithelioid cells (Supplementary Table 4). All MPM patient samples had oncopanel profiling carried out. No tier 1 or tier 2 aberrations which are associated with clinical or potential clinical significance respectively were observed in any of the patient tumor samples (Supplementary Table 5). Therefore, we cannot conclude if chemical vulnerabilities are associated with clinically significant aberrations in the patient samples profiled here. All MPM patient clinical data is available in Supplementary Table 6.

We had the chance to study paired MPM samples from the same patient. One was from a tumor found adherent to the 7th rib (MPS:L) and the other was a tumor from the pleura at a different location. (MPS:M). This gave us an interesting opportunity to ask the question of whether the two distinct but local tumors from the same patient had similar chemical vulnerabilities. The CROCS HTDBP results from both tumors had strong correlation ($r = 0.74$, Fig. 1c and Supplementary Fig. 2B) and 89 of the hits overlapped (MPS:L had 104 hits and MPS:M had 95 hits; Fig. 1d), suggesting that these two distinct tumors had the same chemical vulnerabilities.

### BH3 mimetics navitoclax and S63845 but not venetoclax, enhance apoptotic priming in combination with PI3K/AKT/mTOR pathway inhibitors in primary MPM cells

Representative immunofluorescence microscopy images in a MPM epithelioid (MPS:A), biphasic (MPS:C) and sarcomatoid (MPS:J) patient sample are shown in Fig. 2a and Supplementary Figs. 4–6. These images show the difference between top hits (highest drug-induced priming), non-hits (no drug-induced priming) and DMSO-control under the microscope.

Schematic showing BH3 mimetics and PI3K pathway inhibitors/targets, used in this manuscript are show in Fig. 2b. Two papers

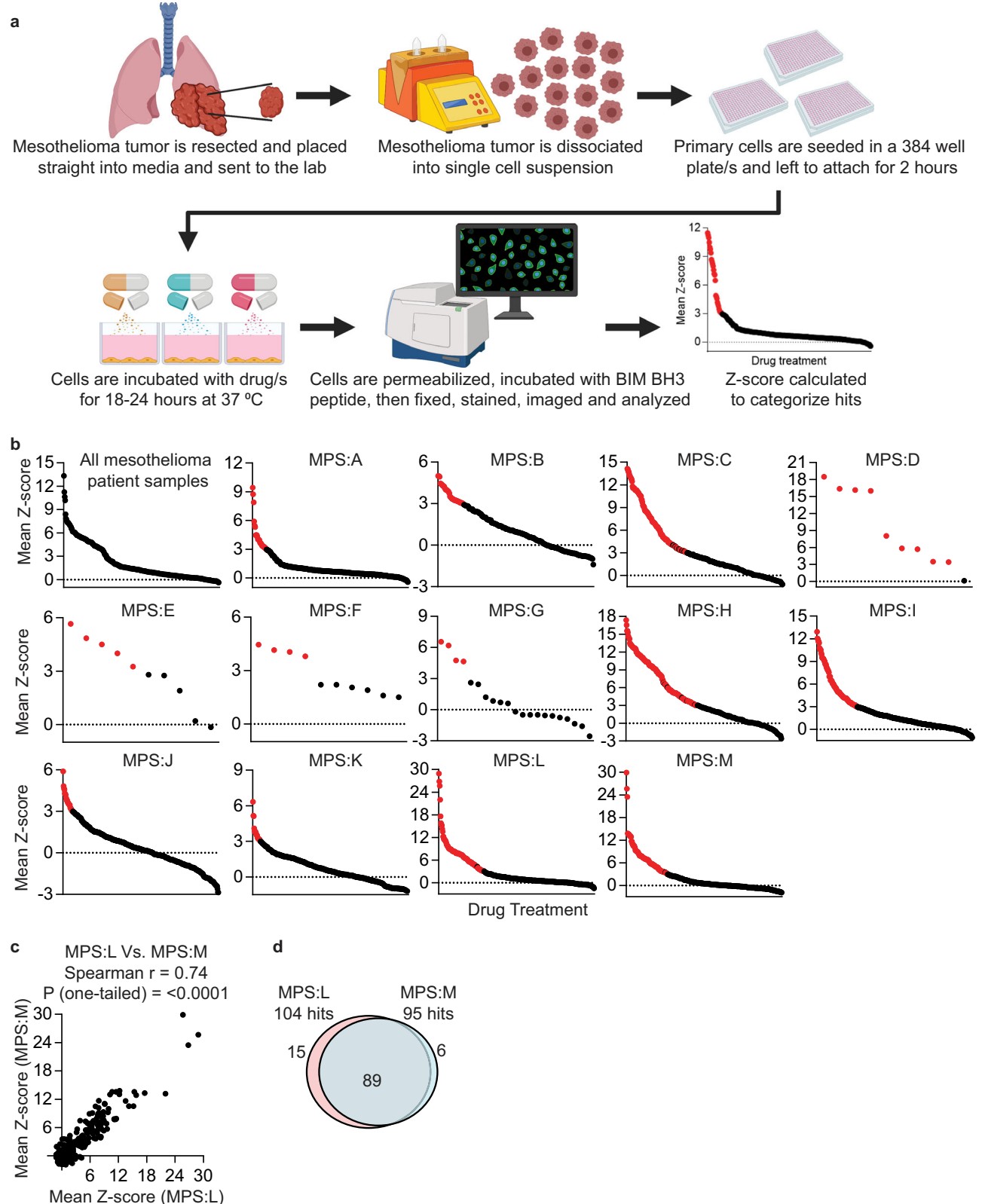

**Nature Communications** | (2023)14:2897

published recently from Arulananda et al. showed the importance of BCL-xL antagonism in the treatment of MPM and how combining with MCL-1 antagonist is highly efficacious in MPM cell lines[26,27]. The most common hits and hits that caused the highest amount of drug-induced priming across all the MPM patient samples are shown in Fig. 2c, Supplementary Tables 7 and 8. BH3 mimetics, specifically navitoclax (BCL-2, BCL-xL, and BCL-w antagonist) and S63845 (MCL-1 antagonist)

in combination with one another, or a PI3K/AKT/mTOR pathway inhibitor, commonly primed primary MPM cells. However, the BH3 mimetic venetoclax (BCL-2 antagonist) did not, suggesting that antagonism of BCL-xL and MCL-1 is more important in primary MPM cells in combination with one another or with PI3/AKT/mTOR pathway inhibitors (Fig. 2d). The entire CROCS HTDBP results for each MPM patient is found in Supplementary Data file 1.

**Fig. 1 | Clinically relevant oncology combination screen (CROCS) on primary MPM patient samples using HTDBP to identify hits.** Fresh primary MPM patient samples were dissociated, treated with CROCS and HTDBP carried out. Cells were analyzed by immunofluorescence microscopy and Z-score was calculated to identify drug/drug combinations that prime tumor cells. **a** Schematic showing the workflow for measuring drug-induced priming using HTDBP on primary MPM patient samples. Created with BioRender.com. **b** Graphs show mean Z-score for each drug treatment (carried out in duplicate), for each primary MPM patient sample (MPS). Each individual dot represents a different drug treatment (single agent or drug-drug combination). A red dot represents a hit with a Z-score ≥ 3 with no replicate <1.5. Black dots are non-hits. **c** Graph showing the mean Z-score correlation between two tumor samples from the same patient (MPS:L and MPS:M) using one-tailed Spearman ranked test (p value = <0.0001). MPS:L is a tumor from the 7th rib and MPS:M is a pleural tumor. **d** Venn diagram showing the overlap between the CROCS HTDBP hits (red dots in part b) for MPS:L and MPS:M.

## Malignant pleural mesothelioma PDX's chemical vulnerabilities overlap with MPM patient samples

To validate HTDBP as an approach to identify efficacious hits we wanted to confirm a top hit that primes the primary MPM cells ex vivo, would also be efficacious in vivo in a MPM PDX model. Before we could start an in vivo efficacy study with a top hit, we wanted to confirm that MPM PDX models recapitulated the pattern of drug sensitivity seen in MPM patient samples. We created three MPM PDX models of biphasic (CPDM_0011x, CPDM_0106x) and sarcomatoid (CPDM_0184x) histological subclasses. While generating MPM PDX's, two epithelioid models developed lymphoma, likely because of the Epstein-Barr virus positivity status of the patient[28]. Therefore, we confirmed each model was to be devoid of lymphoma (hCD45 negative) with lineage match (STR fingerprinting) of the patient tumors, and whole genome copy number profiling to confirm the presence of tumor cells. We then carried out the same approach as in the MPM patient samples in MPM PDX tumors (Supplementary Fig. 7A). Malignant pleural mesothelioma PDX tumors were implanted into the right and left flank of immunocompromised SCID-bg mice. Tumors took between 40 and 120 days to grow enough material for CROCS HTDBP (~1000 mm³; Fig. 3a). All MPM PDX tumors showed good correlation between replicates (Supplementary Figure 7B). Seventy-nine percent of the MPM PDX hits were hits in the MPM patient samples (Fig. 3b and Supplementary Data files 2, 3). Heatmap in Fig. 3c shows all the drug treatments across all MPM PDX tumors profiled and Fig. 3d shows individual Z-score graph for each tumor (red dot is a hit in the assay). The top hits (Fig. 3e, f and Supplementary Tables 9 and 10) were also hits in MPM patient samples, confirming that MPM PDX models demonstrate similar chemical vulnerabilities to primary MPM patient samples. The top two common hits in primary MPM patient (Fig. 2d) and MPM PDX (Fig. 3e) samples were (1) navitoclax plus S63845 and (2) navitoclax plus AZD8055 (mTORC1/2 inhibitor).

We did a tolerance study in non-tumor bearing immunocompromised SCID-bg mice, with navitoclax plus S63845 or navitoclax plus AZD8055, at doses well tolerated as single agents. Unfortunately, all the mice in the navitoclax plus S63845 combination arm died within 4 h of treatment. However, navitoclax plus AZD8055 was well tolerated in vivo (Supplementary Fig. 8A, B). We therefore decided to pursue navitoclax plus AZD8055 to see if this was an efficacious combination in vivo in the MPM PDX model, CPDM_0011x because we had the most data generated for this model (Fig. 3).

## Navitoclax plus AZD8055 is efficacious in vivo in MPM CPDM_0011x PDX model

Mice bearing MPM PDX CPDM_0011x tumors were randomized into 5 arms once tumors reached between 150 and 250 mm³: Arm (1) Vehicle, (2) navitoclax-only, (3) AZD8055-only, (4) navitoclax plus AZD8055 and (5) venetoclax-only. Venetoclax was used as a negative control as it did not cause a significant amount of drug-induced priming in MPM patient and PDX tumors. Both navitoclax and venetoclax were dosed at 100 mg/kg qd for 21 days and AZD8055 was dosed at 16 mg/kg qd for 21 days.

Consistent with HTDBP results, navitoclax plus AZD8055 combination significantly reduced tumor volume compared to all other arms (Fig. 4a; navitoclax plus AZD8055 tumor volume Vs. all other arms = p < 0.0001). Mouse survival was based on time taken to reach the predefined endpoint 5 times initial tumor volume (5xITV). Navitoclax plus AZD8055 increased mouse survival by 30 days from day 19

(vehicle) to day 49 and survival was significantly longer compared to every other arm of the study (Fig. 4b; p < 0.0001). All mice in the vehicle, navitoclax-only, venetoclax-only and 4/7 mice in AZD8055-only, reached the predefined tumor volume endpoint of the study ≤ day 21, indicating the aggressive nature of CPDM_0011x PDX model (Fig. 4c). Relative tumor burden for navitoclax plus AZD8055 combination arm remained significantly low while mice were dosed, compared to all other arms (Fig. 4d). These data determine navitoclax plus AZD8055 combination is efficacious in vivo in CPDM_0011x PDX model, validating HTDBP as an approach to identify efficacious drug combinations ex vivo in MPM.

## AZD8055 reduces MCL-1 protein levels in MPM PDX cells

To investigate the in vivo mechanism of action of the navitoclax plus AZD8055 combination in MPM, CPDM_0011x PDX tumor bearing mice were treated with one dose of either: (1) vehicle, (2) navitoclax-only, (3) AZD8055-only or (4) navitoclax plus AZD8055. Twenty-four hours later tumors were harvested, dissociated, and used for Western blotting analysis. PARP cleavage and caspase 3 cleavage were observed, consistent with an in vivo apoptotic mechanism of tumor cell death. MCL-1 is a known resistance biomarker for navitoclax treatment[29] and AZD8055 has been previously shown to reduce MCL-1 protein levels[30]. AZD8055-only decreased MCL-1 protein levels compared to untreated in the CPDM_0011x PDX cells but had no effect on the apoptotic biomarkers cleaved PARP and cleaved caspase 3 suggesting a reduction in MCL-1 levels alone is not sufficient to induce apoptosis in this model (Fig. 4e). Note that the navitoclax plus AZD8055 combination arm demonstrated preserved MCL-1 levels at the 24-h time point. This is possibly because tumor cells that most downregulated MCL-1 via inhibition of mTOR pathway (AZD8055) in combination with navitoclax have undergone apoptosis (high levels of cleaved PARP and cleaved caspase 3, Fig. 4e) resulting in removal of dead cells from the tumor by phagocytosis. Phospho-S6 is used as a pharmacodynamic biomarker for AZD8055 activity and levels are reduced in arms treated with AZD8055 (Fig. 4e).

## Drug-induced mitochondrial priming is recapitulated in vivo

Our identification of the navitoclax plus AZD8055 combination was predicated on their ability to induce apoptotic priming ex vivo. To investigate whether drug-induced priming also occurred in vivo, we treated mice bearing CPDM_0011x PDX tumors with either vehicle, navitoclax-only, AZD8055-only, or navitoclax plus AZD8055. Twenty-four hours later we carried out flow cytometry based iBH3 profiling[31] on dissociated tumors. Cells were permeabilized and incubated with a range of BIM BH3 peptide concentrations for 1 h, then fixed and stained for cytochrome c. Cytochrome c release was plotted against BIM dose response and BIM AUC (area under curve) and BIM EC50 was calculated to determine relative drug-induced priming. The higher the AUC and lower the BIM EC50, the more primed the tumor cells are because they released more cytochrome c in response to BIM BH3 peptide. All drug treatments significantly primed CPDM_0011x cells in vivo compared to vehicle control (Fig. 4f and Supplementary Table 11). Notable, navitoclax-only and in combination with AZD8055 caused 50% and 44% cytochrome c release, respectively, even without BIM BH3 peptide added (UnT x-axis) in CPDM_0011x PDX cells (Fig. 4f), This suggests that at this the 24-h time point, the effect of the

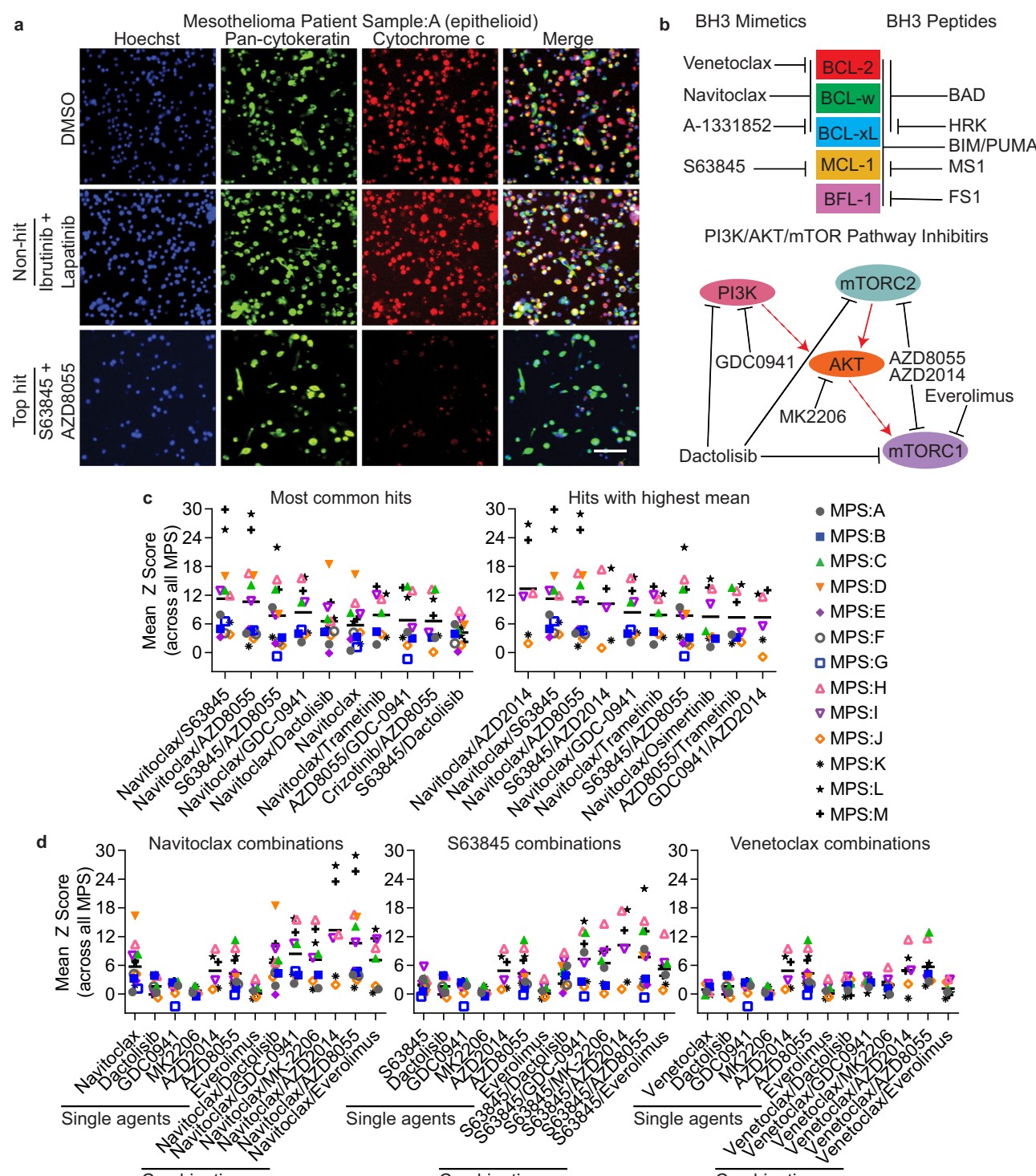

**Fig. 2 | BCL-xL and MCL-1 antagonism enhance priming with PI3K/AKT/mTOR pathway inhibitors in MPM. a** Primary epithelioid MPM cells were treated as previously described in Fig. 1, in technical duplicate. Representative immuno-fluorescence microscopy images from MPM patient sample A (MPS:A). Images taken at tenfold magnification. Hoechst 33342 used to stain DNA (blue) and identify the number of cells/well. Pan-cytokeratin-488 antibody (green) used to identify epithelioid cells (parent population). From the parent population, cytochrome c positive cells % was determined using cytochrome c-647 antibody (red). DMSO treatment is a negative control for cytochrome c loss. Non-hit is a drug treatment that didn't score above the hit threshold (no drug-induced priming). Top hit is the drug treatment that scored the highest mean Z-score (highest drug-induced priming) for this patient sample, first red dot in Fig. 1b. Scale bar is 100 μm. **b** Schematic showing drug targets for BH3 mimetics, BH3 peptides and PI3K/AKT/mTOR pathway inhibitors used in this paper. All drugs (not BH3 peptides) shown here are included in CROCS list except A-1331852 (because it's not in clinical trials yet). BH3 peptides are peptides derived from the BH3 domain of the pro-apoptotic BH3-only Bcl-2 family members and are used in BH3 profiling/DBP assay. **c** Graph showing mean Z-score for top ten most common hits and hits with highest mean across n = 13 MPM patient samples. **d** Graph showing mean Z-score for navitoclax, S63845 or venetoclax in combination with PI3K/AKT/mTOR pathway inhibitors (combinations and single agents), across n = 13 MPM patient samples.

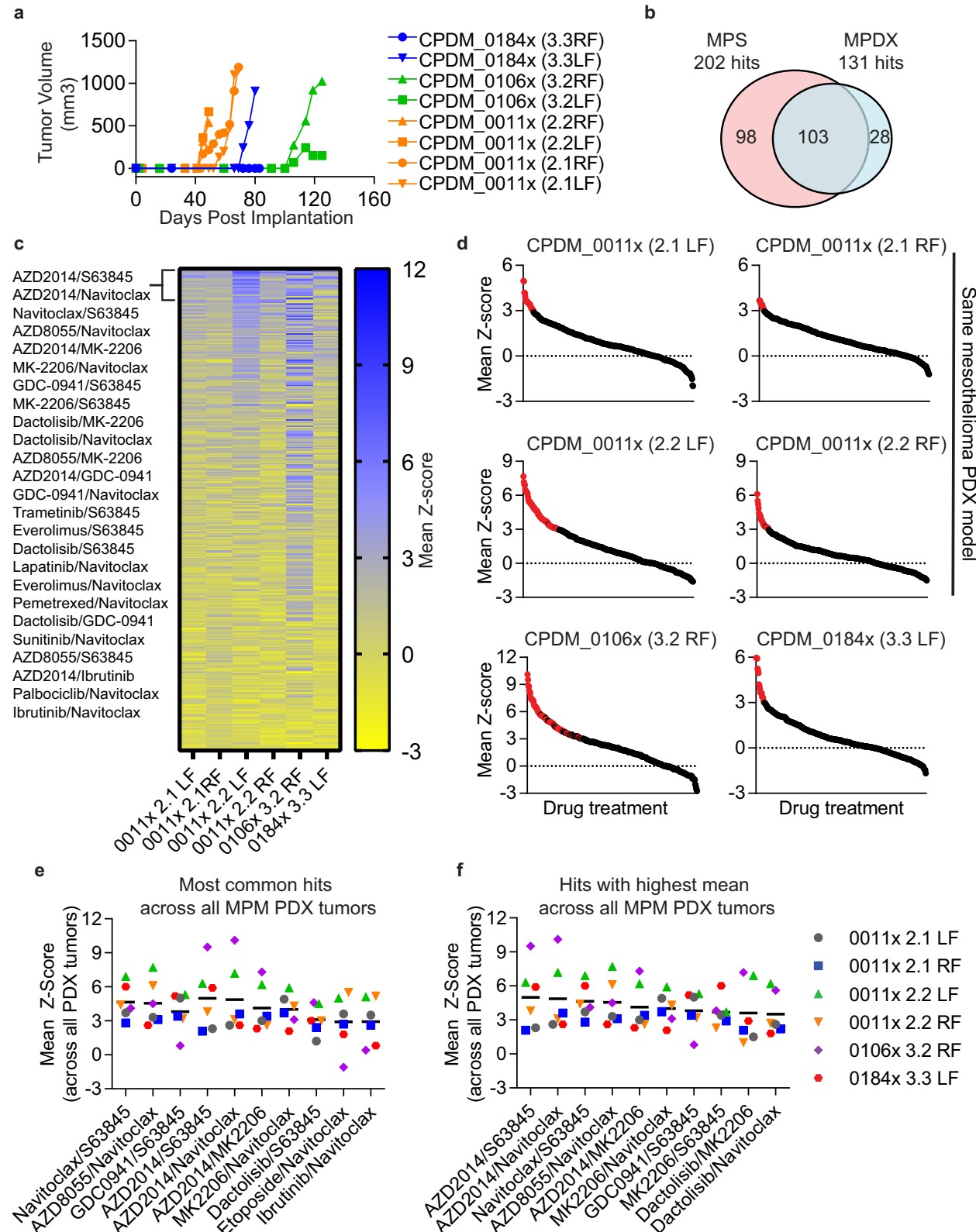

treatment has gone beyond priming and apoptosis had been activated, consistent with immunoblot results in Fig. 4e. Note that single agent navitoclax was nonetheless lacking in efficacy in vivo, likely due to the heterogenous response of CPDM_0011x tumors to navitoclax as a single agent (primed 2/4 of CPDM_0011x tumors ex vivo), highlighting the need for combinations to overcome resistance to single agent navitoclax.

## BCL-xL antagonism drives MCL-1 dependency and AZD8055 drives mitochondrial sensitivity to the BAD BH3 peptide in MPM cell lines

To gain more detailed insight into the molecular mechanism behind the efficacy of navitoclax plus AZD8055, we investigated anti-apoptotic dependencies after treatment with navitoclax or AZD8055 at 24-h, in a panel of MPM cell lines (H2052, JMN, JMN1B and

**Fig. 3 | Clinically relevant oncology combination screen (CROCS) on MPM PDX tumors using HTDBP to identify hits.** Malignant pleural mesothelioma PDX tumors were dissociated, treated with CROCS and HTDBP carried out. Cells were analyzed as previously described in Fig. 1. **a** Malignant pleural mesothelioma PDX tumors (CPDM_0011x, CPDM_0106x and CPDM_0184x) were implanted sub-cutaneously in the right and left flank of immunocompromised SCID-bg mice. When the right (RF) and/or left flank (LF) tumor reached 1000 mm3 mice were sacrificed and tumors harvested for CROCS HTDBP. **b** Venn diagram show the overlap between the CROCS HTDBP hits for MPM patient and PDX samples.

**c** Heatmap showing ranked (highest at the top) Z-score across MPM PDX tumors harvested in (**a**), for each drug treatment. Blue represents a hit with a Z-score ≥ 3 and yellow are non-hits. **d** Graphs show mean Z-score for each drug treatment, for individual MPM PDX tumor harvested in (**a**). One tumor for CPDM_0106x and CPDM_0184x models and four tumors for CPDM_0011x model. Each individual dot represents a different drug treatment (single agent or drug-drug combination). A red dot represents a hit with a Z-score ≥ 3 with no replicate <1.5. Black dots are non-hits. **e** Graph showing mean Z-score for top ten most common hits and **f** hits with highest mean across $n = 6$ MPM PDX tumors.

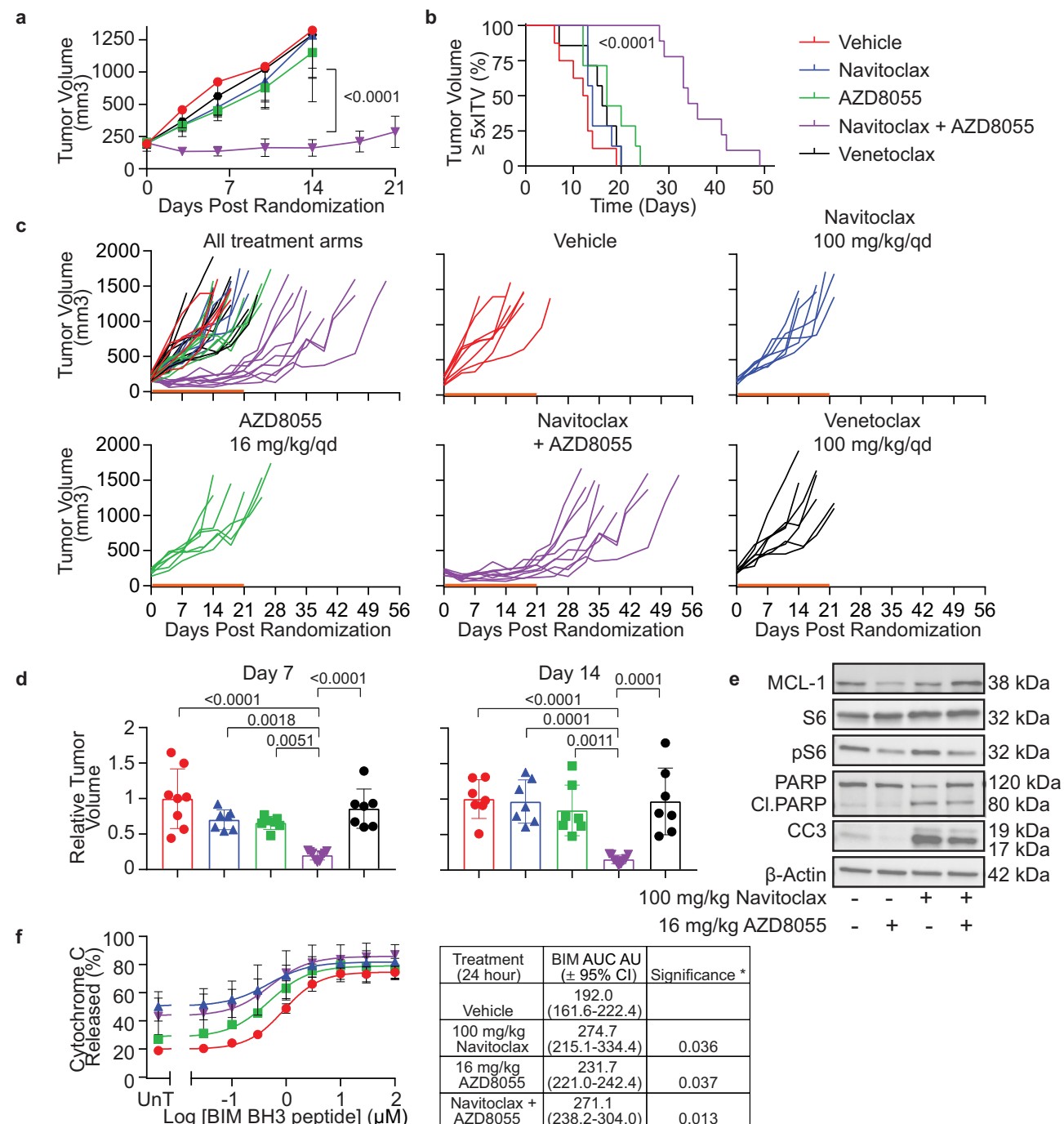

MSTO-211H). We hypothesized that navitoclax treatment would increase MPM cells dependency to MCL-1 when BCL-xL is antagonized, because BCL-xL and MCL-1 are the dominant anti-apoptotic members and compensate for one another in MPM[26,27]. To validate changes are due to BCL-xL and not BCL-2 antagonism, we also treated

cells with single agent A-1331852 (selective BCL-xL antagonist) or venetoclax (selective BCL-2 antagonist). We used DBP to identify anti-apoptotic dependencies at baseline and changes after 24-h drug treatment. Cells were permeabilized and incubated with a panel of BH3 peptides, including sensitizer BH3 peptides which show

**Fig. 4 | In vivo efficacy study validates HTDBP as an approach to identify efficacious hits in MPM.** Malignant pleural mesothelioma PDX model CPDM_0011x, was implanted subcutaneously in the right flank of immunocompromised mice and when tumors reached 150–250 mm³, randomized into 5 treatment groups: vehicle (red, $n = 8$ mice), 100 mg/kg/qd navitoclax (blue, $n = 7$ mice), 16 mg/kg/qd AZD8055 (green, $n = 7$ mice), 100 mg/kg/qd navitoclax plus 16 mg/kg/qd AZD8055 (purple, $n = 9$ mice), or 100 mg/kg/qd venetoclax (black, $n = 7$ mice). Mice dosed for 21 days and sacrificed when tumors reached the endpoint of 5 times the initial tumor volume (5xITV) (**a**–**d**). Mice bearing CPDM_0011x was administered with one dose of indicated drug treatment and 24 h later tumors were harvested and dissociated and used for western blotting analysis (**e**) or iBH3 profiling to measure mitochondrial priming (**f**). **a** Graph showing mean tumor growth over 21 days of dosing. Error bars represent ± standard deviation for 7–9 mice per treatment group. *P* value is <0.0001 for navitoclax plus AZD8055 combination treatment compared to all other treatment groups using a 2-way ANOVA multiple comparisons test. **b** Kaplan–Meier survival curve, mice were sacrificed at 5xITV endpoint. Navitoclax plus AZD8055 combination treatment is statistically significant compared to every other treatment group (*p* value is <0.0001) based on a Log-rank (Mantle–Cox) test. **c** Graph showing the individual tumor volume for each mouse for each treatment group over time. Orange line indicates the dosing period (21 days). **d** Bar graph showing relative tumor burden calculated for each treatment group relative to the mean tumor volume of the vehicle-control group on day 7 and day 14 respectively. Data represents the mean and shows corresponding data points with error bars ± standard deviation. Significance calculated using 2-way ANOVA multiple comparisons. *P* values stated on the graph. **e** Representative immunoblots ($n = 3$) of cell lysates of CPDM_0011x tumor cells after 24-h treatment in vivo for MCL-1, S6, phosphoSer235/236-S6 (pS6), PARP, cleaved PARP (Cl.PARP), cleaved caspase 3 (CC3) and β-Actin (loading control). **f** Intracellular BH3 profiling was performed on CPDM_0011x tumor cells after 24 h treatment in vivo. Cells analyzed by flow cytometry for pan-cytokeratin/vimentin positive cells (parent population) and cytochrome c positive cells %. Graphs represent the mean of three mice ± standard deviation. Table represent BIM area under curve (AUC) ± 95% confidence intervals (CI) and significance is determined according to one-tailed unpaired *t* test comparing vehicle control to treatment. *$P < 0.05$ was considered statistically significant.

specificity in their binding to anti-apoptotic family members and therefore reveal anti-apoptotic dependencies (Fig. 2b). None of the MPM cell lines showed individual anti-apoptotic dependencies at baseline (red bar DMSO-control; Fig. 5a). However, navitoclax and A-1331852 showed significant increased response to MS1 BH3 peptide (MCL-1 antagonist[32]) indicating an increased dependency to MCL-1. Venetoclax treatment had no effect on MS1 BH3 peptide (Supplementary Fig. 9), suggesting that BCL-2 antagonism doesn't affect MCL-1 dependency in MPM. Treatment with S63845 increased anti-apoptotic dependency to BCL-xL (HRK BH3 peptide; Supplementary Figure 9) suggesting that MCL-1 antagonism drives BCL-xL dependency.

Navitoclax mimics the BAD BH3 peptide in that it antagonizes BCL-2, BCL-xL, and BCL-W. We hypothesized that efficacy of the combination might result from a sensitization of mitochondria to the BAD BH3 peptide by AZD8055. Dynamic BH3 profiling revealed that AZD8055 treatment (24-h) significantly increased the response of all the MPM cell lines to the BAD BH3 peptide (Fig. 5b). These experiments highlight the importance of BCL-xL and MCL-1 in MPM for survival, antagonizing one, directly or indirectly, results in increased dependence on the other.

## Navitoclax plus AZD8055 is efficacious in vitro recapitulating efficacy in vivo in MPM

Having observed mechanisms that might explain the effectiveness of the combination in cell lines, we next confirmed that we could recapitulate the in vivo efficacy of navitoclax plus AZD8055, in vitro in these MPM cell lines. After optimizing dosing and timing (Supplementary Fig. 10 and Supplementary Table 12), MPM cell lines were treated for 72 h with either DMSO-control, 1 μM navitoclax, 30 nM AZD8055 or 1 μM navitoclax plus 30 nM AZD8055 combination. Compounds were washed off 3 days later, and colonies were left to grow until day 14. Then colonies were simultaneously fixed and stained, and area confluency and growth rate were calculated (Fig. 5c). All MPM cell lines significantly reduced area confluency and growth rate in the combination compared to either drug as a single agent and DMSO-control. These data confirm that navitoclax plus AZD8055 is efficacious in vitro in the panel of MPM cell lines used to investigate this mechanism.

## Bcl-2 family members levels after treatment with AZD8055 and/or navitoclax in mesothelioma cell lines

There is precedent for the PI3K/AKT/mTOR pathway affecting several Bcl-2 family members including BAD, BIM, BCL-2, MCL-1 and BAX[33]. Previously we showed that MCL-1 protein levels were reduced in MPM PDX cells in vivo after AZD8055 treatment (Fig. 4e).

The effect of navitoclax and/or AZD8055 treatment on a panel of Bcl-2 family members was assessed by Western blotting analysis (protein) and qPCR (mRNA) in MPM cell lines. This data showed some changes common across many of the MPM cell lines, but there was heterogeneity in protein changes (Fig. 5d and Supplementary Fig. 11). We used phospho-AKT as a pharmacodynamic biomarker for mTOR pathway activity. MCL-1 protein levels significantly decreased after treatment with 1 μM AZD8055 in JMN, JMN1B and MSTO-211H cells consistent with what we observed in vivo. BIM levels were significantly increased after all treatment group compared to DMSO-control in H2052, JMN and MST-211H, and significantly increased in the combination group for JMN1B cells. The observed increase in BIM protein levels, offer a molecular explanation for the increase in drug-induced priming after treatment with navitoclax or AZD8055 measured by DBP in MPM cell lines (Fig. 5a, b, 0.3 μM BIM BH3 peptide). Overall, there were not any significant, common changes in relative mRNA levels across the MPM cell line panel after treatment, except for BCL-2 mRNA levels which increased significantly in JMN, JMN1B and MSTO-211H cells for some or all the treatment groups. This translated to a significant increase in protein levels in JMN and MSTO-211H cells but not JMN1B. H2052 cells did show a significant increase in BCL-2 protein levels even though no increase was observed at the mRNA level (Fig. 5d and Supplementary Fig. 11).

## Bcl-xL antagonism sensitizes to the MCL-1 inhibitor triptolide in MPM cell lines

Several other drugs have been shown to downregulate MCL-1. Wei et al. identified the natural compound, triptolide, as a repressor of MCL-1 expression[34] and Busacca et al. showed in MSTO-211H that ganetespib (HSP90 inhibitor) reduces MCL-1 levels[35]. Therefore, we wanted to assess the effect of BCL-xL antagonism with navitoclax or A-1331852 on these MCL-1 repressing compounds. We tested both compounds in H2052 and MSTO-211H cells and confirmed that triptolide nearly completely reduced MCL-1 levels at 24 h, however, ganetespib did not at the concentrations we used (Supplementary Fig. 12A, B). The discrepancy with ganetespib in MSTO-211H cells might be due to the time points used, 24 h used here Vs. 48 h used in Busacca et al.[35]. Based on these data we hypothesized that triptolide would significantly sensitize MPM cell lines to navitoclax or A-1331852 since MCL-1 were reduced so dramatically, while ganetespib would not. BCL-xL antagonism significantly sensitized to triptolide in all 4 MPM cell lines and ganetespib did not, except for H2052 cells (Supplementary Fig. 12C, D). Confirming that BCL-xL antagonism sensitizes to a variety of compounds that antagonize and/or reduce MCL-1 (S63845, AZD8055 and triptolide).

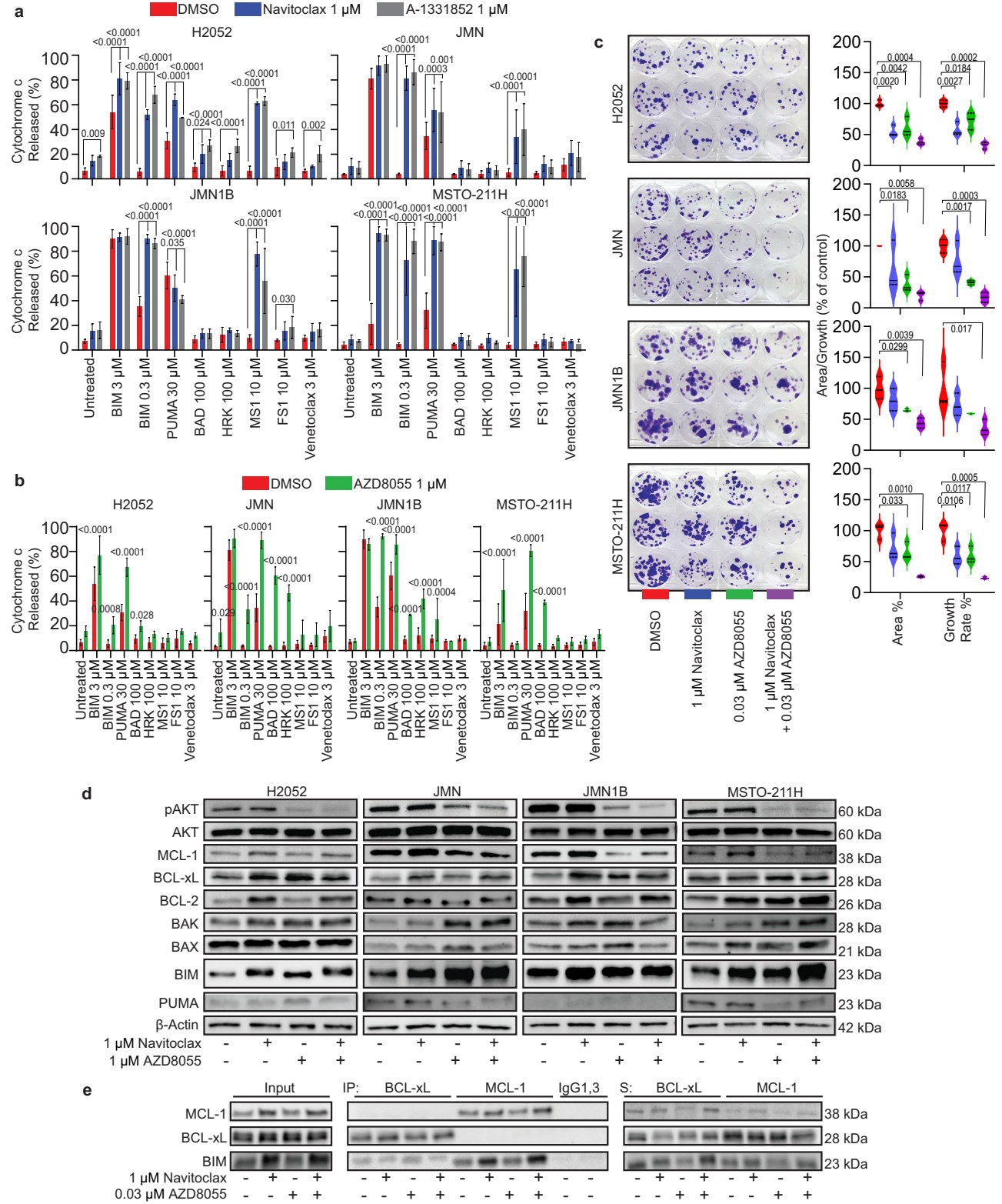

## Navitoclax treatment increased abundance of complexes of MCL-1 and BIM and decreases abundance of complexes of BCL-XL and BIM

As previously described, BCL-xL and MCL-1 are the dominant anti-apoptotic proteins maintaining survival in MPM. We hypothesized that BIM bound to BCL-xL would be removed by navitoclax treatment and MCL-1 would bind and neutralize this newly available BIM. To directly compare the changes in BIM complexes with BCL-xL or MCL-1 after

navitoclax treatment (single agent and combination), we performed immunoprecipitation of BCL-xL and MCL-1 and assessed BIM protein levels in H2052 and JMN1B cell lines. BIM was found in complexes with BCL-xL and MCL-1 at baseline but when cells were treated with navitoclax either as a single agent or in the combination there was a relative reduction in BIM in complexes with BCL-xL and an increase in BIM in complexes with MCL-1, consistent with our hypothesis (Fig. 5e and Supplementary Fig. 13A, B).

**Fig. 5 | Understanding the molecular mechanism of navitoclax plus AZD8055 combination in vitro in MPM.** Analysis of drug-induced priming and anti-apoptotic dependencies after treatment with navitoclax, AZD8055 and a BCL-xL specific antagonist, A-1331852 in MPM cell lines (H2052, JMN, JMN1B and MSTO-211H). Overall priming is measured by BIM or PUMA, whereas HRK, MS1, FS1 and venetoclax are specific for BCL-xL, MCL-1, BFL-1, and BCL-2 dependency respectively (**a**, **b**). Drug treatment with navitoclax, AZD8055 and navitoclax plus AZD8055 combination in MPM cell lines (**c**–**e**). **a**, **b** Cells treated with 1 μM navitoclax, A-1331852 or AZD8055 and then DBP carried out (*n* = 3). Cytochrome c positive cells % (cytochrome c released = 100 − cytochrome c positive cells %) was measured using immunofluorescence microscopy, on permeabilized cells after 1 h incubation with indicated BH3 peptide concentration. Data are presented as bar graphs as mean values with error bars (±standard deviation). We calculated significance using a two-way ANOVA multiple comparisons test to DMSO-control (*n* = 3). **c** Cells

treated with indicated concentration of drug/s for 3 days. Fourteen days from initial drug treatment cells were fixed and stained with crystal violet and area confluency and growth rate was calculated. Data presented as violin plots with median as solid line and quartiles as dashed line. We calculated significance using ANOVA multiple comparisons test to DMSO-control (*n* = 3). **a**–**c** *P* values are shown on the graph. **d** Representative Immunoblots (*n* = 3) of MPM cell line lysates after 24-h treatment with indicated drug concentrations for, phosphoSer473-AKT (pAKT), AKT, MCL-1, BCL-xL, BCL-2, BAK, BAX, BIM, PUMA, and β-Actin (loading control). **e** 24 h after treatment with indicated drug concentration, BCL-xL and MCL-1 were immuno-precipitated in H2052 cells (*n* = 2) and BIM complexes were determined by western blotting analysis (Input total cell lysate; IP, immunoprecipitated fraction; IgG1 (immunoglobulin isotype 1; MCL-1 isotype) and IgG3 (immunoglobulin isotype 3; BCL-xL isotype) control; S, supernatant).

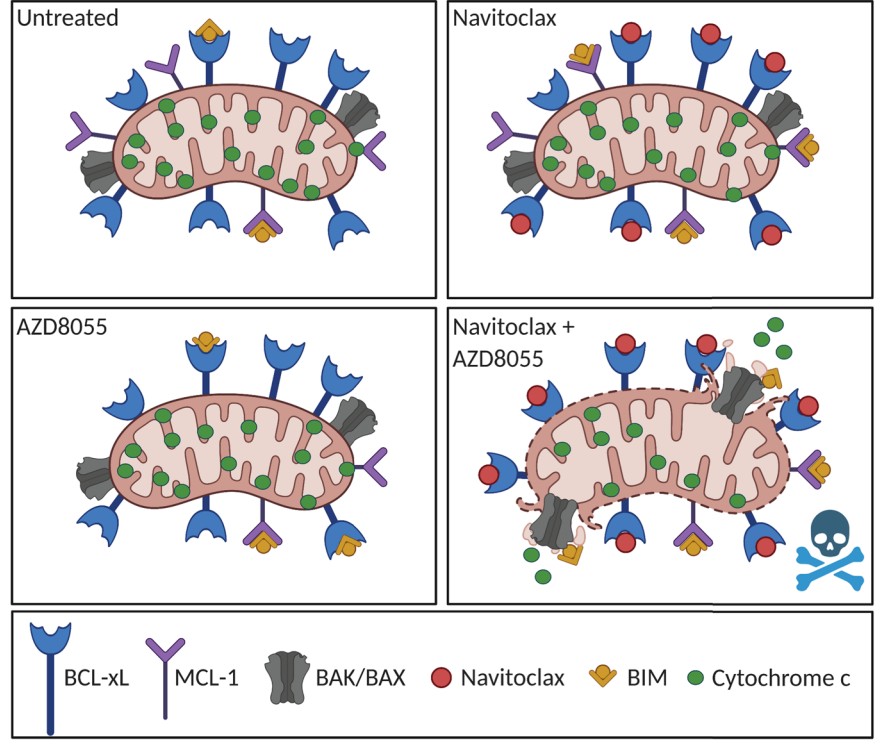

**Fig. 6 | Schematic showing navitoclax plus AZD8055 combination mechanism in MPM.** Malignant pleural mesothelioma cells are relatively "unprimed" at baseline (untreated). No one anti-apoptotic dependency is observed, but a combination of BCL-xL and MCL-1 (BIM bound to both anti-apoptotic's). No MOMP (mitochondrial outer membrane permeabilization) occurs. When cells are treated with navitoclax, this antagonizes BCL-xL. Increased BIM protein levels are observed along with increased binding of BIM to MCL-1. Cells become more MCL-1 dependent. MCL-1 neutralizes the increased BIM pro-apoptotic signal and no MOMP occurs. When

cells are treated with AZD8055, levels of MCL-1 are decreased, and BIM levels increase. BIM is neutralized/bound to both BCL-xL and MCL-1 and no MOMP occurs. When cells are treated with both navitoclax and AZD8055 in combination, BCL-xL is antagonized by navitoclax, MCL-1 levels are decreased by AZD8055 and BIM levels are increased resulting in less BIM bound to BCL-xL, more BIM bound to MCL-1 and remaining BIM activates effectors BAK/BAX, resulting in MOMP. Created with BioRender.com.

## Discussion

The combined antagonism of BCL-xL and MCL-1 is a very attractive therapeutic strategy. Recently, many have shown that targeting BCL-xL and MCL-1 in combination, is highly efficacious in variety of solid disease settings such as cervical cancer, MPM, melanoma and non-small cell lung cancer[26,27,36–38]. A major challenge for dual BCL-xL/MCL-1 inhibition is how to safely dose and maintain tumor efficacy in the mouse and, more importantly, in the patient setting. Combining BCL-xL with MCL-1 inhibitors at therapeutic doses causes liver hepatoxicities in mice[38]. We showed that at therapeutic and well tolerated single agent doses, all mice died within 4 h of navitoclax (100 mg/kg) and S63845 (25 mg/kg) combination. Scheduling/alternating doses of BCL-xL and MCL-1 inhibitors is a way to overcome toxicities associated with directly combining the two drugs. Mukherjee et al., showed that

scheduling of navitoclax and S63845 is possible. They dosed navitoclax at 10 mg/kg (2 days a week) and S63845 at 25 mg/kg (2 days a week), and if the drugs are not dosed on the same day this is well tolerated in mice[39]. Similarly, we have found that toxicities associated with combined MCL-1 and BCL-2 antagonism in mice can be overcome while maintaining efficacy by altering dose and schedule[40]. An alternative to direct antagonism of MCL-1 is to indirectly target MCL-1 through inhibition of pathways that regulate MCL-1 levels, in combination with BCL-xL antagonist.

The efficacy of navitoclax and AZD8055 seen in MPM, is likely due to the antagonism of BCL-xL directly with navitoclax and the indirect downregulation of MCL-1 through AZD8055 in conjunction with upregulation of pro-apoptotic protein BIM (Fig. 6; mechanism schematic). There are two known mechanisms by which MCL-1 could be

indirectly downregulated by mTORC1/2 inhibitor AZD8055. First, AKT is activated by PDPK1 and mTOCRC2, active AKT indirectly stabilizes MCL-1 via phosphorylation and inactivation of GSK3β[41]. GSK3β is a serine/threonine kinase that phosphorylates MCL-1 which causes it to be targeted for degradation so when AKT is dephosphorylated by AZD8055 this results in activation of GSK3β and degradation of MCL-1[42]. Secondly, the mTOR pathway regulates protein synthesis specifically mRNA translation. mTORC1 indirectly activates ribosomal protein S6 and active S6 stimulates the translation of mRNA containing a 5"polypyrimidine tract[43]. Another important effector of mTORC1 is 4EBP1, a repressor of CAP-dependent translation. Active mTORC1 phosphorylates 4EBP1 which prepares 4EBP1 for further phosphorylation, leading to the release of the CAP-binding protein eIF4E and subsequent assembly of the eIF4E/eIF4G complex required for translation initiation[44,45]. Inhibition of mTOR pathway would therefore reduce mRNA translation, selectively affecting proteins with short half-life like MCL-1.

Increased BIM protein levels after navitoclax, AZD8055 and combination treatment implies an increase in pro-apoptotic signaling confirmed by increased drug-induced mitochondrial priming (Fig. 5a, b). AKT negatively regulates members of the forkhead family of transcription factors such as FOXO1 (Forkhead box protein O1) and FOXO3A that promote transcription of BIM[46–48]. Therefore. AKT inhibition can lead to an increase in BIM protein levels.

Others have shown in multiple disease models, including small cell lung cancer and lymphoma, that navitoclax in combination with mTOR inhibitors is efficacious[30,49]. Faber et al. showed that BRAF and KRAS mutant colorectal cancer cell lines were sensitive to navitoclax in combination with AZD8055, but wild type were not[50]. However, Petigny-Lechartier et al. showed that although AZD8055 reduces MCL-1/BH3-only protein ratio by modulating MCL-1 and PUMA, AZD8055 does not efficiently sensitize ovarian cancer cells to ABT-737 (BCL-2, BCL-xL and BCL-w antagonist)[51]. This suggest that the efficacy of combining navitoclax with AZD8055 is disease dependent and being able to identify patients that would respond to the combination is valuable. While this combination had the greatest efficacy in our assay, we still would expect heterogeneity of response even within the MPM population.

We look forward to testing whether DBP can be used to prospectively identify responding patients and we highlight the importance of this combination in MPM potentially as a second-line therapy. We hope to test this hypothesis more formally in a subsequent clinical trial. These studies relied on tissue from resections of MPM, as resections are common practice in this disease. In other tumor types, resection tissue may be more limited, and provided mainly by needle biopsies. We hope to report soon our work on performing experiments like those in this paper using far fewer input cells via a modification of the technique employed here.

## Methods

### Patient samples
Fresh primary MPM tumors obtained from tumor resections, after patients signed an informed consent approved by the Institutional Review Board (Brigham and Women's Hospital and Dana-Farber Cancer Institute protocol 98-063), were used for tumor dissociation to prepare a viable single cell suspension for CROCS HTDBP. Thirteen patient's tumor samples were used in this study.

### Generation of MPM PDX models
The three MPM PDX models are biphasic (CPDM_0011x, CPDM_0106x) and sarcomatoid (CPDM_0184x) histological subtypes. They were created via subcutaneous implantation of tumor specimens collected from consented MPM patients under approved BWH-DFCI protocol 98-063 into NSG mice (NOD.Cg-Prkdcscid Il2rgtm1Wjl/SzJ; The Jackson Laboratory: 005557). Tumors reaching ≥1000 mm³ were harvested and

passaged for at least one additional passage. The established MPM PDX models were screened for the presence of lymphomagenesis[28] and patient lineage is confirmed via STR fingerprinting (Promega GenePrint® 10 System). Low-pass WGS sequencing was performed on the PDX models to confirm the presence of copy number alterations in tumors.

### Cell culture and drugs
H2052 and MSTO-211H cell lines were from ATCC. JMN and JMN1B cell lines were a kind gift from Dr. Raphael Bueno (Brigham and Women's Hospital, Boston MA, USA). All mesothelioma cells were cultured in RPMI media with 10% FBS, 100 U/ml penicillin and 100 μg streptomycin (Gibco) and were incubated in a humidified atmosphere at 37 °C with 5% $CO_2$. Cell lines were authenticated via STR fingerprinting (Promega GenePrint® 10 System; authenticated at Dana-Farber Cancer Institute (DFCI), Molecular Biology Core Facility). JMN and JMN1B STR profiles are not available for reference. Fresh primary MPM tumor cells were cultured in RPMI media with 10% FBS, for no more than 24 h. All drugs used in the clinically relevant oncology combination screen (CROCS) were dissolved in DMSO (10 mmol/L; Sigma) and stored at −20 °C. Drug source is listed in Supplementary Table 1. A-1331852 (MedChemExpress), were dissolved in DMSO (10 mmol/L) and stored at −20 °C. For in vivo use navitoclax, S63845 and venetoclax powder was stored at −20 °C. Navitoclax was then formulated in 10% DMSO, 30% polyethylene glycol 400 (Sigma) and 60% Phosal 50 PG (American Lecithin Company) and stored at room temperature (RT) for up to 7 days. Venetoclax was formulated in 10% ethanol, 30% polyethylene glycol 400 and 60% Phosal 50 PG and stored at 4 °C for up to 5 days. S63845 was formulated in 25 mM hydrochloric acid and 20% hydroxylpropyl-beta-cyclodextrin (Sigma) and used the same day (within 3 h). AZD8055 was formulated in 30% Captisol® (Sulfobutylether-β-Cyclodextrin; Ligand Technology, San Diego, USA) and stored at 4 °C for up to 5 days.

### Drug treatment and cell viability
Primary MPM cells were seeded at 5000 cells per well in RPMI + 10% FBS media of a 384-well plate. Cell Titer Glo assay was used to assay viability of primary MPM cells over 24-h. Cells were drugged using the HP D300e digital dispenser (Hewlett-Packard) at 1 μM for each drug as a single agent or in the clinically relevant oncology combination screen (CROCS) and high-throughput dynamic BH3 profiling (HTDBP) was carried out 18–24 h later. The full list of CROCS drugs is shown in Supplementary Table 1. Malignant pleural mesothelioma cell lines were treated with 1 μM AZD8055, navitoclax, A-1331852, S63845 or venetoclax for 24 h and then DBP was carried out.

Malignant pleural mesothelioma cell lines were treated with AZD8055 or BH3 mimetic dose response (30–0.1 μM, half log fold decrease in concentration) plus the indicted concentration of combination drug or DMSO-control (single agent drug) for 24 or 72-h. Cell viability was measured using Cell Titer Glo assay according to manufacturer's instructions (Promega). Luminescence relative to untreated cells (no drug) was determined, relative to each drug dose response or dose response with indicated drug combination. We calculate AUC (area under curve) for individual drug dose response (DMSO-control) or combination using GraphPad Prism (version 8). Statistical analysis was carried out on three independent AUC readings. To determine drug $EC_{50}$, log drug concentration was plotter against raw luminescence, and nonlinear curve fit analysis was performed (GraphPad Prism). Statistical analysis was carried out on three independent $EC_{50}$ readings.

### Colony formation assay
Malignant pleural mesothelioma cell lines were seeded at 500 cells per well in a 12-well plate and allowed to settle overnight and treated as indicated in the figure legends. Cells were washed after 72-h drug

treatment, and medium was replaced with fresh medium (no drug/s). Fourteen days after initial drug treatment, cells were fixed and stained simultaneously with 0.05% (w/v) crystal violet, 1% formaldehyde and 1% methanol (Sigma) in 1× PBS (Gibco) and washed 3 times in $H_2O$. Area confluency of colonies was counted using image J colony area plugin[52]. Relative growth rate was assessed by solubilizing cells in 10% (v/v) acetic acid and measuring absorbance at 590 nm.

## Tumor dissociation

A scalpel was used to cut patient or PDX tumors into small pieces (<2 mm thick) and then tumor pieces were incubated in RPMI containing 10 mg/mL collagenase IV, 650 U/mL DNase I and 500 U/mL hyaluronidase (Roche Diagnostics) for 1 h at 37 °C with occasional mechanically dissociation using Miltenyi gentleMACS Dissociator. Dissociated cells were filtered through a 70 μm cell strain and viability was assessed[23].

## DBP and HTDBP, imaging and data analysis

Dynamic BH3 profiling was carried on MPM cell lines seeded at 1000 cells/well in 384-well plate and drug treated the next day using the D300e digital drug dispenser (Hewlett Packard). Freshly dissociated primary MPM and MPM PDX cells were seeded at 5000 cells/well in collagen type 1 (Sigma #C7661) coated (20 μg/$CM^2$) 384-well plate and 2 h later cells were treated with CROCS. Media/drugs were washed from plates using the BioTek 406EL plate washer (BioTek) and replaced with PBS. A 2× BH3 profiling buffer (1× is 150 nM Mannitol, 10 mM HEPES-KOH pH 7.5, 50 mM KCL, 0.02 mM EDTA and EGTA, 0.1% BSA and 5 mM Succinate, Sigma) was added to cells with a final digitonin (Sigma #D5628) concentration of 0.002%. BH3 peptide were added using the D300e digital drug dispenser at appropriate concentrations. 1 h later cells were fixed in paraformaldehyde and neutralized in Tris/Glycine buffer (1.7 M Tris, 1.25 Glycine pH 9.1, Sigma). Cells were stained with antibodies in staining solution (10% BSA, 2% Tween20 in PBS, Sigma) overnight and washed with BioTek plate washer the next day and imaged[22]. All imaging was performed on the ImageXpress Micro Confocal High-Content Microscope (Molecular Devices; at the ICCB-Longwood screening facility at Harvard Medical School). A 10× objective was used and multi wavelength cell scoring was performed to analyze images using MetaXpress (Molecular Devices; at the ICCB-Longwood screening facility at Harvard Medical School). The adaptive background correction module was used to segment cells based on intensity above the local background resulting in single cell segmentation and area of pan-cytokeratin or cytochrome c staining intensity. Cells are scored as negative or positive based on the area[23]. All subsequent analysis was carried out in Excel or GraphPad Prism to generate delta priming and Z-score described in detail in the statistical analysis and reproducibility section of methods.

## Antibodies for DBP/HTDBP and western blotting

For DBP and HTDBP, nuclei were stained with Hoechst 33342 (1:2000; Life Technologies) to determine total number of cells. A pan-cytokeratin antibody (#628608; 1:1000; Biolegend) was used to identify epithelioid MPM tumor cells parent population and vimentin (#677809; 1:1000; Biolegend) was used to identify sarcomatoid MPM parent population. Cytochrome c positive cells % was measured using cytochrome c-Alexa647 antibody (#612310; 1:2000; Biolegend). Western blotting was performed per manufacturer specifications (BIO-RAD). For Western blotting all antibodies were from Cell signaling Technology unless otherwise stated, β-Actin (#4967S 1:2000), AKT (#4685S 1:1000), phosphoSer473-AKT (#4685S 1:1000), BIM (#2933S 1:1000), BAK (#12105S 1:1000), BAX (#2772S 1:1000), BCL-xL (#2764S 1:1000), BCL-2 (#15071S 1:1000), cleaved Caspase 3 (#9661S 1:1000), MCL-1 (#39224S 1:1000; human specific for PDX samples), MCL-1

(#94296S 1;1000), PUMA (#4976S 1;1000), PARP (#9542S 1;1000), S6 (#2317S 1;1000) and phosphoSer235/236-S6 (#4857S 1;1000).

## Tumor cell lysis and immunoprecipitation assay

Cells were lysed using ice cold CHAPS lysis buffer (1% CHAPS, 20 mM TRIS-HCL pH 7.5, 137 mM NaCl, 5 mM $MgCl_2$, 1 mM EDTA and 1 mM EGTA), protease inhibitor cocktail (P8340), phosphatase inhibitor cocktail 2 (#P5726) and 3 (#P0044) (Sigma) according to manufacturer's instructions. For immunoprecipitation, lysates were incubated overnight at 4 °C with Protein G Dynabeads™ (Invitrogen) and the following antibodies: anti-MCL-1 (12 μg; BD Pharmingen), anti-BCL-xL (8 μg; EMD Millipore), and anti-mouse IgG1 (12 μg; #5415S) and IgG3 (8 μg; #37988 S) isotype control (Cell Signaling Technology). The immunoprecipitate (IP) was divided equally to blot for MCL-1, BCL-xL, and BIM. Twenty micrograms of the supernatant, and the initial whole-cell lysate for each condition (input) was loaded onto the gel with the IP and analyzed by Western blot analysis (as previously described).

## Intracellular BH3 profiling (iBH3) and CROCS HTDBP on mesothelioma PDX cells

All mice were maintained within the DCFI animal facility and all experiments involving animals were conducted in accordance with the DFCI policy and animal protocol, reviewed and approved by the DFCI Institutional Animal Care and Use Committee. Tumor size cannot exceed 2 cm in any direction according to this protocol and maximal tumor size was not exceeded. Mice were euthanized by $CO_2$ inhalation when they met the predefined study endpoint mentioned, or when tumors reached 2 cm in any direction, or when body weight dropped below 15% (whichever comes first). Mice were housed in vented caging systems in a 12-h light/12-h dark environment and maintained at uniform temperature and humidity. Malignant pleural mesothelioma PDX models CPDM_0011x, CPDM_0106x and CPDM_0184x, were grown by subcutaneous injection of tiny tumor chunks into the mid-dorsal flank of 8-week-old female SCID-beige mice (C.B-17/IcrHsd-PrkdcscidLystbg-J; Envigo). For CROCS HTDBP, mice were implanted in the right and left flank and when one of the tumors reached 1000 mm³ the mouse was sacrificed. Tumors were harvested and CROCS HTDBP carried out as previously described. For iBH3 profiling when CPDM_0011x tumors implanted in the right flank reached ~700 mm³, mice were randomized into 4 groups of 3 mice per group: (1) vehicle, (2) 100 mg/kg navitoclax, (3) 16 mg/kg AZD8055 or (4) 16 mg/kg AZD8055 and 1 h later, 100 mg/kg navitoclax. Twenty-four hours after drug/s were administered, mice were sacrificed, and tumors harvested. Tumors were dissociated as previously described. Same antibodies used in DBP/HTBP were used here to stain MPM PDX cells. Then iBH3 was carried out as previously described[31]. Cells were analyzed by flow cytometry (BD Fortessa analyzer) using DIVA software (BD Biosciences) Gating strategy was SSC-A Vs. FSC-A (all cells); SSC-W Vs. SSC-H (singlets); Hoechst Vs. SSC-A (live cells); Pan-cytokeratin-488 Vs. SSC-A (epithelioid parent population); Cytochrome c Vs. SSC-A (cytochrome c +ve cells).

## In vivo tolerance of drug combinations

Three 8-week-old female SCID-beige mice were housed for each treatment arm as previously described. Mice were treated for 21 days by oral gavage with either 100 mg/kg navitoclax, 100 mg/kg venetoclax, 16 mg/kg AZD8055, or 16 mg/kg AZD8055 and 1 h later, 100 mg/kg navitoclax. S63845 was given by intravenous tail injection at 25 mg/kg twice a week for 21 days. For the navitoclax and S63845 tolerance study, mice were dosed with navitoclax 100 mg/kg by oral gavage and then 8 h later, S63845 at 25 mg/kg by intravenous tail injection. For the duration of the dosing, mice were monitored twice daily for any changes in weight and body appearance. Monitoring was continued for a further two weeks after which the animals were sacrificed.

### In vivo efficacy of BH3 mimetic and etoposide

CPDM_0011x MPM PDX tumors were grown in SCID-beige mice and housed (5 mice to a cage) as previously described. Mice were monitored twice weekly for signs of tumor growth. Once a palpable tumor was detected, measurements were taken twice a week with calipers. Tumor volume was calculated using the formula $0.5 \times$ (longest measurement) $\times$ (shortest measurement)$^2$. Once tumors measured between 150–250 mm$^3$ they were randomized, using the deterministic method, into 5 groups (7–9 mice per group): (1) vehicle, (2) 100 mg/kg navitoclax, (3) 16 mg/kg AZD8055, (4) 16 mg/kg AZD8055 and 1 h later 100 mg/kg navitoclax, or (5) 100 mg/kg venetoclax. Navitoclax, AZD8055 and venetoclax were administered by oral gavage daily for 21 days. Tumor measurements were continued three times a week until the tumor reached five times initial tumor volume (5xITV) after which the mouse was sacrificed.

### Statistical analysis and reproducibility

For CROCS HTDBP, delta priming % (=(% positive cytochrome c cells mean DMSO-control-treated wells) − % positive cytochrome c cells drug-treated well), and Z-score (=(% negative cytochrome c cells drug-treated well − (% negative cytochrome c cells mean DMSO-control-treated wells)) / (standard deviation DMSO-control-treated wells)), were calculated in Excel for each drug treatment well in all 13 MPM patient samples and 3 MPM PDX models (4 tumors for CPDM_001x, 1 tumor for CPDM_0106x and 1 tumor for CPDM_0184x). All CROCS treatments were carried out in duplicate for each tumor. Delta priming % correlation was analyzed using one-tailed Spearman Rank correlation in GraphPad Prism and $p < 0.05$ was considered statistically significant. A CROCS HTDBP hit is defined as a mean Z-score ≥ 3 with no Z-score replicate <1.5. Statistical analysis for in vivo efficacy study tumor volume and tumor burden was carried out using a two-way ANOVA multiple comparison, comparing the means from each group in GraphPad Prism. Comparison of in vivo efficacy study survival curves used the Log-rank (Mantle–Cox) test in GraphPad Prism. $P < 0.05$ was considered statistically significant for all in vivo statistical analysis. BIM BH3 peptide dose response AUC was calculated in GraphPad Prism and statistical analysis was carried out using unpaired, one-tailed t tests to compare treated and control groups. T tests were performed in Excel (Microsoft) to determine significance. $P < 0.05$ was considered statistically significant. Statistical analysis for in vitro DBP was carried out using a two-way ANOVA multiple comparison, comparing each cell mean with every other cell mean of that row in GraphPad Prism. $P < 0.05$ was considered statistically significant.

### Reporting summary

Further information on research design is available in the Nature Portfolio Reporting Summary linked to this article.

## Data availability

All data needed to evaluate the conclusions in the paper are present in the article, source data and supplementary files. Source data are provided with this paper.

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

## Acknowledgements

We thank Peter Sorger, Laura Maliszewski and Jennifer Smith for support at the Laboratory of Systems Pharmacology at Harvard Medical School. We thank Jeremy Ryan for making BH3 peptides and Patrick Bhola for advice on using HTDBP. A.L. acknowledges support from R35 CA242427, Ludwig Cancer Research at Harvard and the Starr Cancer Consortium. K.L.L., K.H.C., and R.B. report funding from DOD CA160891. R.B. acknowledges support from the National Cancer Institute RO1CA120528.

## Author contributions
D.S.P. and A.L. designed the study and wrote the manuscript. D.S.P. performed all experiments. R.D. and S.B. helped DSP with in vivo and vitro experiments. R.B. provided advice on experimental design and R.B., K.L.L. and K.H.C. helped edit manuscript. R.B. provided mesothelioma patient samples. R.B., K.L.L. and K.H.C. created MPM PDX models and collected MPM patient data.

## Competing interests
A.L. discloses consulting and sponsored research agreements with AbbVie, Novartis, and AstraZeneca. He is on the scientific advisory board of Anji Onco, Flash Therapeutics, Dialectic Therapeutics, and Zentalis Therapeutics. K.L.L. discloses consulting from BMS, Integragen, and Rarecyte. D.S.P. is now employed by AstraZeneca. K.L.L. is a founder and equity holder of Travera LLC. K.L.L. has sponsored research agreements with BMS and Lilly. The following are US Patents regarding BH3 profiling, owned by Dana-Farber: 10,393,733; 9,902,759; 9,856,303; 9,540,674; 8,221,966; 7,868,133. A.L. is an inventor on patent applications US20180128813A1, US20180120297A1 held/submitted by Dana-Farber Cancer Institute that covers HTDBP. Remaining authors declare no competing interests.

## Ethics
Primary MPM tumors described in the manuscript were obtained, after patients signed an informed consent approved by the Institutional Review Board (#98-063). All mice used in the manuscript were maintained within the DCFI animal facility and all experiments involving animals were conducted in accordance with the DFCI policy and animal protocol, reviewed and approved by the DFCI Institutional Animal Care and Use Committee.
