## [Peer Review File · Nature Communications]

Reviewers' Comments:

Reviewer #1:

Remarks to the Author:

General comments

The authors describe the use of a technique known as dynamic BH3 profiling to identify combinations of treatment that could readily engage intrinsic apoptosis. The experimental evidence is supported by ex-vivo primary mesothelioma (dissociated tumour), in vivo mouse data (PDX) and cell lines and identifies AZD8055 and navitoclax as an optimal proapoptotic combination. BH3 profiling has shown itself to be predictive for clinical responses in other treatment settings. However in this present study, clinical data is not yet available. The authors propose that this combination should however be explored based on the data presented.

Overall this is an interesting , novel and generally well executed body of work focusing on a rare cancer of unmet need.

Expression of BCLXL and MCL1 in mesothelioma have been previously reported (as opposed to BCL2) and strategies that coordinately target these proteins would be expected to unleash BAX/BAK and activate mitochondrial apoptosis (Willis et al, Genes Dev 2005). The findings are therefore not unexpected and should apply to other cancers where these pro-survival BCL2 family proteins predominate.

This strategy if applied with BH3 mimetics is highly toxic however (as shown), and the authors although touching on this, should discuss in more detail strategies to overcome this potential issue if/when taking this into the clinic. I assume no phase 1 data exists for the navitoclax/AZD8055 combination.

The paper, although examining intra-tumour heterogeneity in one patient, does not discuss the challenge of inter patient heterogeneity, with the overriding implication that all patients should respond. I think this is unrealistic and more discussion is required to consider how best to target this treatment.

Specific comments

The authors suggest that mesothelioma may be curable. Is mesothelioma curable at all, or are some patients destined to have highly indolent disease based on their genetic make up eg. germline bap1 ? I suggest that the term cure is removed or replaced with reference to "long-term survivors". The implication otherwise is that treatment alone is responsible for complete elimination of the cancer for which evidence is largely lacking.

51. For advanced disease ipilimumab and nivolumab is approved for 1st line therapy based on the Checkmate 743 pivotal phase III trial.

68. please provide a reference for BOX

73. I think the definition of priming requires further clarity for non-experts. Is it correct to say that priming is the result of constitutive, endogenous occupancy of prosurvival BCL2 family by sensitizer- BH3 only proteins? Some sort of statement to this effect would be helpful.

91. the number of patient's studied should be mentioned in the results.

109-123 This initial section of the results details the methods and should be moved to the methods section section I think.

116. seeded

123. why was duplicate and not triplicate used (to allow a statistical measure of variance such as

SD). This level of technical replicate would be considered standard in other settings.

132 It is interesting that inter-tumour heterogeneity was examined. one tumour is described as being on the 7th rib, the other pleural. Presumably both were pleural in origin. Perhaps the location/origin of the second tumour could be given for consistency?

151 BCLXL antagonism in the.....

163. "...confirmed to devoid of lymphagenesis issue..." should be re-written for clarity

179 was the cause of death established. If known this should be highlighted. Why was this schedule of treatment lethal but MCL1 downregulation by AZD8055 not -is the MCL1 downregulation seen, tumour specific ?

211 This sentence should read AZD8055 alone decreased....

213 if AZD8055 downregulates MCL1 as its proapoptotic mode of action, but cells remained with preserved MCL1, does must implicate some intra-tumour heterogeneity that could account for resistance if these cells that retain MCL1 can persist and are clonogenic. What might account for these persistent cells – failure to downregulate MCL1?

253 please correct to – “showed significantly increased response”

Mesothelioma is a very heterogeneous cancer. Sarcomatoid tumours exhibit profound chemotherapy resistance (see checkmate 743 phase III trial, Baas et al, Lancet 2021), likely due to epithelial mesenchymal transition. Were there any differences between histological subtype correlates with mitochondrial priming?

364 three PDX models were represented by two biphasic and one sarcomatoid histological subtypes. However, the epithelioid subtype is by far the most common type of mesothelioma, and accounts for 70%, so its not clear why at least one of the models were not representative.

A number of other drugs tested in mesotheliomas have been shown to be able to downregulate MCL1 -including epirubicin,(Wei et al, Cancer Cell 2012) and HSP90 inhibition (Busacca et al, Oncogene 2015, Tong et al, Mol Cancer ther 2017). Would these alternative drug combinations be as likely to achieve the same activity as AZD8055 or S63485 and if not, why not?

265 Do the authors feel that the results demonstrating adaptive dependencies on BCLXL or MCL1 could be exploited by sequential versus concurrent treatment. This seems to be the implication eg. priming by navitoclax might drive dependency to an MCL1 inhibition by AZD8055. Patients might risk less toxicity? I think its important to discuss the implications of this experiment in any case.

277 investigate this mechanism

280 Do you mean precedent rather than precedence ?

288 please clarify the significance of increased BAX/BAK expression. Was this expected and is this transcriptional? My understanding is that these oligomerize to activate.

313 Do you mean sequencing rather than scheduling? This paragraph nicely discusses the implications mentioned above

320 Is indirect targeting of MCL1 via AZD8055 likely to be tumour specific?

324 this sentence seems to be incomplete or should be re-written “through AZD8055 in the setting of upregulation of pro-apoptotic proteins” could read, “...of MCL1 by AZD8055 in conjunction with upregulation of....”

Based on the molecular mechanisms implicated in this study, do the authors feel that all

mesotheliomas should respond to navitoclax + AZD8055, or as is likely, some or most tumours will demonstrate resistance intrinsic resistance. The subject of tumour heterogeneity is not discussed. How might a tumour evade this combination?

Mesotheliomas commonly exhibit inactivation of the hippo pathway which can activate TEAD induced BCLXL, survivin and BMF transcription. Another very common mutation BAP1, alters calcium homeostasis/mitochondrial cytochrome C release and therefore apoptosis threshold. Do the authors feel that inter-patient genomic heterogeneity could modify responses to navitoclax+AZD8055? It is interesting that at 348 the paper cited by Petigny-Lachartier showed resistance in wild type BRAF/KRAS. Genetic context therefore matters (albeit in another cancer in this example). This should be discussed in more detail. The implication from this work is that mesotheliomas are homogeneous and would all respond to this navitoclax/AZD8055.

Some mesotheliomas in TCGA exhibit MCL1 amplification, or FBXW7 deletion which might be expected to increase MCL1 expression. Might this help or hinder this therapeutic strategy?

Reviewer #2:

Remarks to the Author:

In the manuscript entitled "Dynamic BH3 profiling identifies novel combinations in malignant pleural mesothelioma with in vivo efficacy", by Potter et al., the authors perform high throughput dynamic BH3 profiling to test the effect of many drugs and drug combinations in patient mesothelioma samples ex vivo. The authors identify mTORC1/2 inhibition by AZD8055 in combination with BCL-xL/BCL-2/BCL-w inhibitor Navitoclax as a highly active ex vivo treatment combination, which is followed up with in vivo validation in mouse, and mechanistic studies pointing to AZD8055 downregulating MCL-1 expression and increasing dependence on BCL-xL, thereby explaining the combined effect of these two drugs.

Taken together this is a timely manuscript that provides exciting insights into a possible treatment avenue for a hard to treat cancer, mesothelioma. The authors deserve praise for their efforts in this direction. However, the study as presented has shortcomings that will need to be addressed.

Major points:

Fig 1:

Fig 1b could benefit from a different visualization that allows better comparison of the measurements between samples. For example, a heatmap of samples x drugs could be helpful, perhaps just for those drugs measured in most (or all) samples, and for the top-hits identified in each sample, to see how they reproduce across samples.

The authors should clarify - perhaps in the form of flow chart or table - which compounds were measured across which samples, as well as how and why certain compounds and combinations were prioritized for testing. This would make it easier to understand how the combinations for the in vivo follow-up were selected.

For example: Given that the samples on which only around 10 drugs were tested already contained the hits chosen to follow-up on, it is not entirely clear if the authors already knew the hit drug combinations they wanted to follow-up on starting their screen.

Another example: It appears the newer mTORC1/2 drug AZD2014 was only tested on later (higher letter) samples, but then still included in 'priority' panel of ~10 drugs tested on those samples with little material. Why? Did the authors already pick their hits at that stage and still screened more samples?

Figures 1 and 3 suggest quite some patient sample variability. It would be good if the authors could analyze in more depth the clinical features associated with the patient samples which they screened, and their relationship to sensitivity to BCLi/mTORCi combinations. Were some samples perhaps obtained from patients that had received treatments already, whereas others may have been treatment-naive? Are there pathology or genetic-level differences between these samples?

This might explain in part observed patient variability that the plots suggest but not address, and greatly add to the novelty and interpretation of the presented work.

For the combination screen drug treatment, the authors should clarify whether combinations were 1uM total concentration or 1 uM per drug. If it is 1uM for each drug, the authors should test if the effect they see for the combination is equal to the additive effect of each single drug.

Fig 2:

Fig 2a and S3-5: Top hits consistently reduce the cell number - how does the BH3 priming z-score correlate with reduction in cell number across all screens? Is the BH3 readout necessary at all to identify these top hits?

Fig 2c and d: The visualization is difficult to read, maybe the plots could be replaced by dotplots / heatmaps to show the data per sample, and the mean z-score could be indicated as a barplot on top.

Relating also to the flow questions on Figure 1: Panel d could be swapped with panel c, since I was initially quite confused how the drugs shown in panel c were chosen, and it made more sense after looking at panel d.

Fig 3:

Fig 3c: same comment as Fig. 1b, a drug response matrix/heatmap (even just of selected drugs) would greatly improve the readability of the per-sample drug responses the authors measure.

Fig 3d: same comment as Fig 2c/d.

Fig 4:

Fig 4f: The difference in the BIM peptide AUCs seem to come mostly from differences in baseline priming without BIM peptide added, as BIM EC50 is comparable across conditions. Can you then conclude that treated cells are more primed? Is baseline apoptosis the same as priming?

Fig 5:

Fig 5 a-c: The number of replicates used for ANOVA is not specified

Fig 5 c: It is not clear which statistical comparisons are being made between the bars, as the placement of stars is ambiguous.

Fig 5 d: The text specifies that AZD8055 increases BAK in 4/4 cell lines, and BAX in ¾ cell lines, however the differences are in some cases small, and could be affected by differences in loading (e.g. with MSTO-211H). For some cell lines, the increase also appears to be reversed by co-treatment with Navitoclax. Similarly, it is unclear whether BIM increases with treatment in JMN1B cells as the bands are overexposed. Given the small effect size, I would therefore ask that the authors provide further replicates with a quantification of band intensity relative to loading control to allow for a more robust evaluation of these differences.

Fig 5e: Could the differences in detected BIM be due instead to differences in total levels of MCL-1 and BCL-xL present (due to changes in expression or differences in loading)? For example, in figure S10, BIM levels appear to be elevated by Navitoclax in both the IP and S fraction. Further information on how the Western blots were loaded and performed should be provided in the materials and methods.

Fig 6:

AZD8055 likely has many more effects than downregulation of MCL1. To see if it really is MCL1 downregulation that causes the combined drug effect, it would good to do a knockdown of MCL1, and/or to perform transcriptomics on AZD8055 drug treated cells, to show that MCL1 downregulation is a key feature and most likely mechanistic explanation for the observed drug

effect. If, in the end, the drug effect is merely additive, not synergistic, it is the interpretation on the mechanistic synergy proposed could be reduced as it is not critical for the study.

Materials and methods:

The "Tumour cell lysis and Immunoprecipitation Assay" requires more detail on how the western blot was conducted. Was the entire IP and supernatant loaded in each condition, or were samples normalised to total protein content etc.?

Minor comments

Writing comments / typos etc:

The manuscript feels a bit rough, including typos, and small mistakes in figures and citations. Below is a non-exhaustive list, it would be great if the authors could go over their manuscript carefully to double-check and correct such issues.

(L28) "in a MPM" -> "in an MPM"

(L77) "anti-apoptotic's" -> "anti-apoptotics"

(L107) Please keep the interpretation on "increased ATP production", not "increased viability".

(L108) "RMPI" → "RPMI"

(L164) "to devoid" → "to be devoid"

(L188+) "mg/kg/qd" -> "mg/kg/d" or "mg/kg qd"

(L503) "standard deviated" -> "standard deviation"

Axis label in Fig 4b: "Tumor volume > 5xITV" -> "Tumor volume < 5xITV"

References: Reference 21 and 23 are the same paper.

Please rephrase Figure S1 legend: "A BIM dose response on untreated primary MPM tumor cells was carried 16 hours after seeding, to calculate the optimum BIM BH3 peptide concentration. The point where MOMP is about to occur (BIM EC10) and drug treatments that prime MPM cells would readily release cytochrome c (surrogate for priming), after being incubated with optimum BIM BH3 peptide concentration."

Please rephrase Figure S3-S5 legend: "Hoechst 33342 used to stain DNA (blue) and therefore identify the number of cells present in each well. Pan-cytokeratin-488/vimentin-488 antibody (green) used to identify sarcomatoid cells (parent population)."

Reviewer #3:

Remarks to the Author:

In this interesting and thought provoking manuscript the authors examine the utility of high throughput dynamic BH3 profiling for identifying drug combinations that effectively kill mesothelioma in primary cultures and PDX models. The authors have pioneered this technique and should be lauded for moving it towards clinical utility.

The results of the study are very interesting and it appears that the combination of AZD8055 and Navitoclax not only confers 'priming' but that this priming translates to tumor kill in PDX models. Overall I am very enthusiastic about most of the results in the paper. My main problem is with the molecular model proposed to explain the increased efficacy of the combination.

One of the major conclusions from the paper is: "Mechanistic investigation revealed that AZD8055 treatment down regulates MCL-1 protein levels, increases BIM protein levels, and increases MPM mitochondrial dependence on BCL-xL, which is inhibited by navitoclax. Navitoclax treatment increases dependency to MCL-1 and increases BIM protein levels."

These statements are based to a large degree on measurements of BCL-XL, MCL-1 and BIM protein levels by western blotting (data shown in Figure 5D-E and supplementary Figure 10). To my eye in Figure 5D addition of Navitoclax not AZD8055 increased BIM protein in H2052 cells, both Navitoclax and AZD8055 increased BIM in JMN cells, JMN1B the blot is overexposed and therefore uninterpretable, and Navitoclax increased BIM more than AZD8055 increased BIM in MSTO-211H cells. MCL-1 levels were unaffected in H2052 and decreased to varying degrees by AZD8055 in the other cell lines. The patterns are also inconsistent with Figure 4e (see below). The magnitude of changes looks similar for BAK and BAX but those changes are also not correlated across cell lines. The most striking observation is the decrease in pAKT as a result of AZD8055 addition.

In 5e the immunoprecipitates are also difficult to interpret. It appears that Navitoclax increases protein levels for MCL-1 and BCL-XL and that results in more co-precipitation of BIM with MCL-1. This is not a surprising result and I am pretty sure that BIM binding to MCL-1 has been reported to stabilize the proteins. I remind the authors that co-precipitation does not indicate direct binding despite what is claimed in the manuscript. In addition, the cell lysis buffer used contains Triton, a detergent known to artificially promote interactions between Bcl-2 type proteins as elegantly shown by Youle years ago.

Figure S10 shows no Bim binding to Bcl-XL (the size change suggests the band shown is a non-specific band) and even when MCL-1 decreased a lot, it bound the same amount of BIM when cells were treated with the mTor inhibitor as without the mTor inhibitor.

However, since none of this was quantitated, no replicates are provided and only thin slices of blots are presented, interpretation of the western blots is really difficult.

Minor points –

Are all the abbreviations really necessary? The result is phrases like "followed by HTDBP on CROCS-treated primary MPM cells" which in the absence of a glossary are very difficult to follow.

Another highlighted conclusion is that "These findings validate the principle that HTDBP can be used as a functional precision medicine tool to rationally construct combination drug regimens in MPM and other cancers." However, it is not clear how much primary cancer tissue is required. Based on the amount of PDX material required (a tumor diameter of about 1 cm) it would seem that these analyses are not practical for tumors from which the tissue sample would be limited to one or two fine needle biopsy samples. Instead, the analyses are limited to tumors for which tissue is obtained during resection. Since the authors propose that the current experiments with mesothelioma are exemplary for other tumor types the limitations imposed by the tissue requirements should be discussed by the authors.

Figure 1c shows the correlation of the drug responses for two samples from different tumors in the same individual. While the correlation is obvious, it is not clear how correlated the results are between patients for examples between MPS:C and MPS:H or MPS:H with MPS:L

In Figure 4e adding AZD8055 doesn't increase Cl PARP or decrease pS6 or increase CC3 there is just a small increase in Mcl-1 with Navitoclax and a small decrease in MCL-1 without Navitoclax and a decrease in pS6. The authors should comment more on the significance of this data.

In Figure 5a-b In general there isn't a big change with Bim peptide plus AZD8055 across cell lines. The more impressive changes are for PUMA and BAD. Is that due to the differences in peptide concentrations or are the large differences in peptide concentrations reflective of variations in peptide binding? What do the authors make of these differences?

In Figure S9 the decrease in luminescence due to the addition of Navitoclax or A-1331852 to AZD8055 is not very impressive. How does this fit with the model being presented?

Reviewer #4:

Remarks to the Author:

In this paper, the authors apply their “dynamic BH3 profiling” to identify combination therapies for mesothelioma. This method allows them to assess apoptotic priming as a correlate of efficacy, permitting the use of primary patient samples rather than culture adapted cell lines. Here they show the utility of the approach and provide compelling data that the results are highly reproducible across multiple tumors, that primary tumors and PDX tumors behave similarly, and that the results have application to in vivo therapy in preclinical models. The paper is a tour de force of the use of this methodology.

Before I go into details, I feel it is important to state that while the top hit they produced is not especially surprising (it has been known for some time that mTOR inhibition reduces Mcl-1 levels and sensitizes for killing by Navitoclax), the fact that this emerged from dynamic BH3 profiling of primary specimens is exciting and important. It also points to a significant finding that I discuss in more detail below.

My relatively minor questions and concerns follow in no particular order.

1. As mentioned above, the fact that cross-specimen comparisons show high correlations is striking (and exciting), as is the correlation with PDX. While the combinations that represent “top hits” would also be expected to appear in cell lines analyzed in a similar manner, it would be helpful to know the extent to which the same panel of agents and combinations show less correlation between primary samples and cell lines (such as those they used). If so, it would add impetus for the use of primary samples and dynamic BH3 profiling for drug combination, therapeutic discovery.
2. The finding that Bim protein levels are elevated by the treatment is important (and is likely to have mechanistic value). Since Bim is regulated at both transcriptional and translational (stability) levels, it would be helpful to know if the upregulation is at the mRNA level (or not).
3. Perhaps the most important finding is that while the combination of Navitoclax with the Mcl-1 inhibitor S6 is highly lethal in vivo, the combination of the mTor inhibitor with Navitoclax is well tolerated. (The rapidity of death-4 hrs-with the Nav/S6 combination suggests possible cardiotoxicity, which has been described). Of course, Torc1/2 inhibition and BH3-mimetic treatment have different mechanisms of action on Mcl-1; the first relies on the short half-life of Mcl-1. This leads to a fundamental question: does the mTOR inhibitor sensitize for Navitoclax without depleting all Mcl-1 (i.e., is there a stable pool of the protein that protects the animal but not the tumor?). Indeed, the disparate results on Mcl-1 levels in Fig. 5e (not reduced with AZD8055) and Fig. S10 (reduced with AZD8055) are suggestive and should be discussed. If possible, it might be useful to assess levels of Mcl-1 expression and extent of cell death in hearts of animals treated with the two combinations (since we do not know the mechanism of toxicity, this may be asking too much, but it may be worth a look).

Reviewer #1 (Remarks to the Author):

General comments

The authors describe the use of a technique known as dynamic BH3 profiling to identify combinations of treatment that could readily engage intrinsic apoptosis. The experimental evidence is supported by ex-vivo primary mesothelioma (dissociated tumour), in vivo mouse data (PDX) and cell lines and identifies AZD8055 and navitoclax as an optimal proapoptotic combination. BH3 profiling has shown itself to be predictive for clinical responses in other treatment settings. However in this present study, clinical data is not yet available. The authors propose that this combination should however be explored based on the data presented.

Overall this is an interesting , novel and generally well executed body of work focusing on a rare cancer of unmet need.

Expression of BCLXL and MCL1 in mesothelioma have been previously reported (as opposed to BCL2) and strategies that coordinately target these proteins would be expected to unleash BAX/BAK and activate mitochondrial apoptosis (Willis et al, Genes Dev 2005). The findings are therefore not unexpected and should apply to other cancers where these pro-survival BCL2 family proteins predominate.

This strategy if applied with BH3 mimetics is highly toxic however (as shown), and the authors although touching on this, should discuss in more detail strategies to overcome this potential issue if/when taking this into the clinic. I assume no phase 1 data exists for the navitoclax/AZD8055 combination.

Toxicity is a concern with BH3 mimetic combination, therefore an indirect mechanism to down regulate MCL-1 levels seems advantageous if tolerated. The dose of navitoclax and AZD8055 used for the efficacy study was well tolerated based on mouse body weight and appearance

With regards to toxicity to MCL-1 inhibition, this is an ongoing issue that many companies that make MCL-1 inhibitors are currently investigating. It is likely that scheduling will be important at reducing toxicities in BH3 mimetic combinations.

Currently navitoclax in combination with the more clinically relevant mTORC1/2 inhibitor AZD2014 (not AZD8055) is in phase 1/2 study in patients with relapsed SCLC and other solid tumors. No results have been posted yet and the trial is due to end in August 2022.

ClinicalTrials.gov Identifier: NCT03366103. We hope the reviewer agrees that it would be

premature of us to comment on clinical toxicity of this combination, as the true results, provided by others, should be available within the year.

The paper, although examining intra-tumour heterogeneity in one patient, does not discuss the challenge of inter patient heterogeneity, with the overriding implication that all patients should respond. I think this is unrealistic and more discussion is required to consider how best to target this treatment.

We agree, it is naive to think that all MPM patients would respond to navitoclax + AZD8055 combination due to the heterogenous nature of the disease. We chose this combination because it seemed to have the broadest efficacy in the cases that we studied. We have added text to the end of our Discussion section to make this clearer: "While this combination had the greatest efficacy in our assay, we still would expect heterogeneity of response even within the MPM population. We look forward to testing whether DBP can be used to prospectively identify responding patients and we highlight the importance of this combination in MPM potentially as a second line therapy."

Specific comments

The authors suggest that mesothelioma may be curable. Is mesothelioma curable at all, or are some patients destined to have highly indolent disease based on their genetic make up eg. germline bap1 ? I suggest that the term cure is removed or replaced with reference to "longterm survivors". The implication otherwise is that treatment alone is responsible for complete elimination of the cancer for which evidence is largely lacking.

We agree with the reviewer in that mesothelioma is rarely cured and we say that in the manuscript. In fact, the only time we mention cure in the manuscript is when we say few patients are cured (line 45).

51. For advanced disease ipilimumab and nivolumab is approved for 1st line therapy based on the Checkmate 743 pivotal phase III trial.

Changed and citation updated

68. please provide a reference for BOX

Sorry, this was a typo, it should have been BOK not BOX and the reference used referred to BOK.

73. I think the definition of priming requires further clarity for non-experts. Is it correct to say that priming is the result of constitutive, endogenous occupancy of prosurvival BCL2 family by sensitizer- BH3 only proteins? Some sort of statement to this effect would be helpful.

Great point. We have modified the text to give clarity on priming and anti-apoptotic binding availability/occupancy (line 74-78).

91. the number of patient's studied should be mentioned in the results.

Thank you. We have updated text to include sample number which is 13 MPM patient samples.

109-123 This initial section of the results details the methods and should be moved to the methods section section I think.

Thank you. We moved some of this text to the methods and materials section.

116. seeded

Thank you, changed

123. why was duplicate and not triplicate used (to allow a statistical measure of variance such as SD). This level of technical replicate would be considered standard in other settings.

Patient samples are precious, and we are not guaranteed to get a lot of tissue so there is a tradeoff between number of drugs to study and replicates you can do. This is a major limitation when using patient samples. If we reduced the number of drugs in the screen, we could increase the replicates, but we decided to increase the drugs used in the screen and scarified one of the replicates. In at least 4 samples we only had between 170,000- 325,000 cells (Table S2).

132 It is interesting that inter-tumour heterogeneity was examined. one tumour is described as being on the 7th rib, the other pleural. Presumably both were pleural in origin. Perhaps the location/origin of the second tumour could be given for consistency?

Both tumors were of pleural origin, but one was a tumor from the pleura and the other was found adherent on the 7th rib. Text has been updated to give clarity (line 148).

151 BCLXL antagonism in the..... Edited, now line 169.

163. "...confirmed to be devoid of lymphogenesis issue..." should be re-written for clarity

Thanks for bringing this up. While generating novel MPM PDX's, some of the epithelioid models developed lymphoma as they were passaged in vivo. This was likely because the patients were positive for the Epstein - Barr virus and proliferation of lymphoma cells in the immune-compromised environment out-competed the proliferation of MPM cells. Therefore, all models were confirmed to be devoid of lymphomagenesis. Text has been updated for clarity and reference included.

179 was the cause of death established. If known this should be highlighted. Why was this schedule of treatment lethal but MCL1 downregulation by AZD8055 not -is the MCL1 downregulation seen, tumour specific ?

The mice were removed by the vets in the animal facility because they started dying within 1 hour of the second compound being dosed (navitoclax was dosed first and then S63845). All 4 mice were dead within 4 hours. We asked the vet if they did a necropsy, and unfortunately, they didn't. The vet said was it was clear the mice died from severe toxic events. That is all the information we have on toxicity of the combination.

It is unknown why direct MCL-1 antagonism is more toxic than indirect downregulation, but it could be due to the degree of MCL-1 inhibition between S63845 and AZD8055. Or it could be on target or off target toxicity with S63845 that is not observed with AZD8055 because the mechanism of action is different. S63845 directly antagonizes MCL-1, where AZD8055 downregulates MCL-1 protein levels so the mechanism is different.

We think it is the indirect downregulation of MCL-1 that gives navitoclax plus AZD8055 its therapeutic index, whereas navitoclax plus S63845 directly antagonizes MCL-1 and that is clearly toxic. However, we acknowledge this is speculative, and we hope the Reviewer understands that explication of the toxicities of MCL-1 inhibitors is now a significant industrial effort, and beyond the scope of our studies.

211 This sentence should read AZD8055 alone decreased.... Changed

213 if AZD8055 downregulates MCL1 as its proapoptotic mode of action, but cells remained with preserved MCL1, does must implicate some intra-tumour heterogeneity that could account for resistance if these cells that retain MCL1 can persist and are clonogenic. What might account for these persistent cells – failure to downregulate MCL1?

Great question. You can expect some degree of heterogeneity with expression of most proteins across the tumor. In this case it might be likely that any persistent cells might still express MCL-1 at relatively higher levels than more sensitive cells. To address whether a more complete reduction in MCL-1 levels would further sensitize to navitoclax we treated mesothelioma cell lines with a drug that had been shown to reduce levels of MCL-1, triptolide (Wei et al., 2012). Triptolide nearly completely reduced MCL-1 protein levels at 0.1 and 1 μ M (Sup. Figure 12a). Triptolide alone had mild to moderate effect on cell viability in 24-hours but when combined with either navitoclax or the BCL-xL specific antagonist A-1331852, this completely killed all the cells in the combination (Sup. Figure 12d). These results support the idea that reduction of MCL-1 levels is toxic, especially in combination with BCL-XL inhibition, and is consistent with the concept that heterogeneity in MCL-1 reduction could lead to development of resistance.

253 please correct to – “showed significantly increased response” **Changed**

Mesothelioma is a very heterogeneous cancer. Sarcomatoid tumours exhibit profound chemotherapy resistance (see checkmate 743 phase III trial, Baas et al, Lancet 2021), likely due to epithelial mesenchymal transition. Were there any differences between histological subtype correlates with mitochondrial priming?

Unfortunately, we only received one sarcomatoid patient sample and therefore, we cannot confirm any significant differences between mitochondrial priming or top hits between epithelioid and sarcomatoid histologies. Even though biphasic tumors are made up of both epithelioid and sarcomatoid cells we wanted to see if there were any significant differences between epithelioid and biphasic patient samples. There were no significant differences between mitochondrial priming or top hits between epithelioid and biphasic histologies (Table S4). This may be because the presence of epithelioid cells in the biphasic tumors diluted our ability to detect differences of a sarcomatoid subset, or there may truly be little difference to observe.

364 three PDX models were represented by two biphasic and one sarcomatoid histological subtypes. However, the epithelioid subtype is by far the most common type of mesothelioma, and accounts for 70%, so its not clear why at least one of the models were not representative. We agree with this statement. When we started this project there were not any commercially available MPMP PDX's, so decided to try to generate MPM PDX's ourselves. We started out with 6 MPM PDX models that grew in vivo. 3 epithelioid, 2 biphasic and 1 sarcomatoid. Unfortunately, 2/3 of the epithelioid models transformed into lymphoma, likely because the

patient was positive for Epstein - Barr virus (as mentioned above). As the models were passaged the lymphoma cells out competed the MPM cells so those two models were scrapped. The final epithelioid model unfortunately didn't grow after 6 months implanted in the mice, so we had to drop this model too. The only models at that time that grew were the biphasic and sarcomatoid MPM PDX models.

A number of other drugs tested in mesotheliomas have been shown to be able to downregulate MCL1 -including epirubicin,(Wei et al, Cancer Cell 2012) and HSP90 inhibition (Busacca et al, Oncogene 2015, Tong et al, Mol Cancer ther 2017). Would these alternative drug combinations be as likely to achieve the same activity as AZD8055 or S63485 and if not, why not?

Probably the most profound synergistic effect of MCL-1 inhibition was its synergy with Bcl-xL antagonism. Therefore, to address this comment, we investigated whether other compounds known to downregulate MCL-1 would sensitize cells to Bcl-xL antagonists navitoclax and A-1331852. We used triptolide shown by Wei et al., 2012 to downregulate MCL-1 and ganetespib shown by Busacca et al., 2016 to downregulate MCL-1. In our hands Triptolide drastically and significantly reduced MCL-1 levels at 24 hours but ganetespib did not (Sup. Figure 12a-b; see new Figure below). Busacca et al., 2016 showed a reduction in MCL-1 levels after treatment with ganetespib but they treated for 48 hours so this may be a reason for the discrepancy. In this case, therefore, ganetespib treatment acted as a negative control as a perturbation that did not reduce MCL-1 levels. Fitting with this data, triptolide significantly sensitized cells to navitoclax and A-1332852 treatment and ganetespib did not (Sup. Figure 12c-d). These new results agree with the Reviewer's hypothesis that other agents that reduce MCL-1 levels might achieve activity similar to AZD8055 or S63845.

Figure S12: Ganetespib or Triptolide in combination with BCL-xL antagonist in MPM cell lines. Malignant pleural mesothelioma cell lines (H2052, JMN, JMN1B and MSTO-211H) were exposed with the indicated concentration ganetespib or triptolide for 24 hours. (a) Immunoblots of MPM cell lysates after 24-hour treatment with indicated concentration of drug for MCL-1 and β -Tubulin. (b) Graph showing relative MCL-1 levels to DMSO-control for part (a). Significance was based on one-way ANOVA comparing to DMSO-control. Cell lines were treated with a dose response to ganetespib (c) or triptolide (d) for 24-hour in combination with either DMSO-control (ganetespib-only or triptolide-only), 1 μ M Navitoclax or 1 μ M A-1331852. * significance was based on one-tailed unpaired t-test versus ganetespib-only or triptolide-only area under curve (AUC) for the same cell line. Cell viability was assessed using Cell Titer Glo. Graph represents three independent experiments \pm standard deviation.

265 Do the authors feel that the results demonstrating adaptive dependencies on BCLXL or MCL1 could be exploited by sequential versus concurrent treatment. This seems to be the implication eg. priming by navitoclax might drive dependency to an MCL1 inhibition by AZD8055. Patients might risk less toxicity? I think its important to discuss the implications of this experiment in any case.

Potentially yes, we know sequentially dosing navitoclax-break-s63845 is better tolerated (Mukherjee at al., 2020). We have also show ourselves that toxicities associated with combined MCL-1 and BCL-1 antagonism in mice can overcome and maintain efficacy by altering dose and schedule (Bhatt et al., 2020). However, we show the best efficacy is with the combination group compared to sequential or alternating doses between venetoclax and S63845, highlighting the importance of this combination. This manuscript stemmed from the want to identify clinically relevant combinations directly on patient samples. Understanding sequentially dosing is out of the scope of this paper. However, as far as dosing the patient with navitoclax plus S63845 sequentially/alternating to reduce toxicities in MPM, this is important and should be investigated in MPM.

277 investigate this mechanism updated

280 Do you mean precedent rather than precedence ? Precedent- updated

288 please clarify the significance of increased BAX/BAK expression. Was this expected and is this transcriptional? My understanding is that these oligomerize to activate.

Once activated, BAK and BAX form homo/hetero oligomers that form pores in the outer mitochondrial member causing the release of cytochrome c and the induction of apoptosis. Even though BAK and BAX show an increase in all the MPM cell line panel, quantification of the blots showed it was not significant (except for JMN1B BAX levels in AZD8055 treatment) and therefore we have edited the text to remove the statement implying it's significance.

313 Do you mean sequencing rather than scheduling? This paragraph nicely discusses the implications mentioned above

We mean scheduling/alternating doses and edited the text to clarify this. Agreed, this is what we were discussing earlier regarding navitoclax plus S63845 toxicities associated with combining the drugs and how scheduling/alternating dosing is a way to try and mitigate toxicities associated with the combination.

320 Is indirect targeting of MCL1 via AZD8055 likely to be tumour specific?

While mTOR is widely expressed, we suspect that mTOR biology differs across tissues and it is clear that mTOR biology is distinct in malignant tissues. Perhaps at least as relevant is that we have shown in the last decade or so that in response to even equal perturbations, cancer cells tend to be more likely to engage in potentially fatal apoptotic signaling than normal tissues.

324 this sentence seems to be incomplete or should be re-written “through AZD8055 in the setting of upregulation of pro-apoptotic proteins” could read, “...of MCL1 by AZD8055 in conjunction with upregulation of....”

Thank you, updated.

Based on the molecular mechanisms implicated in this study, do the authors feel that all mesotheliomas should respond to navitoclax + AZD8055, or as is likely, some or most tumours will demonstrate resistance intrinsic resistance. The subject of tumour heterogeneity is not discussed. How might a tumour evade this combination?

With regards to navitoclax plus AZD8055 2 out of 13 patients didn't respond to the combination so it is likely that we would observe inter-patient heterogeneity with navitoclax and AZD8055 combination. A likely explanation for intrinsic resistance to this combination is through AZD8055 treatment not regulating MCL-1 protein levels or through upregulation of the anti-apoptotic BFL-1 (Yecies et al., 2010; Thomalla et al., 2022). Tumor heterogeneity is discussed further in the next comment.

Mesotheliomas commonly exhibit inactivation of the hippo pathway which can activate TEAD induced BCLXL, survivin and BMF transcription. Another very common mutation BAP1, alters calcium homeostasis/mitochondrial cytochrome C release and therefore apoptosis threshold.

Do the authors feel that inter-patient genomic heterogeneity could modify responses to navitoclax+AZD8055? It is interesting that at 348 the paper cited by Petigny-Lachartier showed resistance in wild type BRAF/KRAS. Genetic context therefore matters (albeit in another cancer in this example). This should be discussed in more detail.

This is a very interesting and important point about genetic context. It is very likely that inter-patient genomic heterogeneity could modify response to navitoclax/AZD8055 combination. To investigate the inter-patient genomic heterogeneity, we had oncopanel testing carried out on all 13 MPM patient samples. For all the genes included in the oncopanel, none of the patient

samples had any aberrations in tier 1 (strong clinical significance) or tier 2 (potential clinical significance). Tier 3 and 4 variants were observed but these mutations have uncertain clinical significance or function and therefore we cannot draw any conclusions on these findings or make correlations between genetic context and chemical vulnerabilities. The text has been edited to include the oncopanel results for each patient is included in Table S5.

The implication from this work is that mesotheliomas are homogeneous and would all respond to this navitoclax/AZD8055.

It is not our contention that all MPM cases would respond. While we found a consistent apoptotic response to navitoclax and S63845 ex vivo in patient samples, in vivo in MPM PDX samples and in vitro in MPM cell lines, this is not the case for navitoclax and AZD8055. We think MPM would likely be heterogenous in their response to navitoclax and AZD8055. We hope that dynamic BH3 profiling would provide a more individualized predictive biomarker, but that is a clinical hypothesis we have yet to test.

Some mesotheliomas in TCGA exhibit MCL1 amplification, or FBXW7 deletion which might be expected to increase MCL1 expression. Might this help or hinder this therapeutic strategy?

It is difficult to predict, it might depend on whether the MCL-1 is largely occupied or unoccupied by pro-apoptotic proteins. Across many tumors, the association between the expression of an anti-apoptotic protein and sensitivity to an antagonist has proven to be somewhat weak.

Reviewer #2 (Remarks to the Author):

In the manuscript entitled "Dynamic BH3 profiling identifies novel combinations in malignant pleural mesothelioma with in vivo efficacy", by Potter et al., the authors perform high throughput dynamic BH3 profiling to test the effect of many drugs and drug combinations in patient mesothelioma samples ex vivo. The authors identify mTORC1/2 inhibition by AZD8055 in combination with BCL-xL/BCL-2/BCL-w inhibitor Navitoclax as a highly active ex vivo treatment combination, which is followed up with in vivo validation in mouse, and mechanistic studies pointing to AZD8055 downregulating MCL-1 expression and increasing dependence on BCL-xL, thereby explaining the combined effect of these two drugs.

Taken together this is a timely manuscript that provides exciting insights into a possible treatment avenue for a hard to treat cancer, mesothelioma. The authors deserve praise for their efforts in this direction. However, the study as presented has shortcomings that will need to be addressed.

Major points:

Fig 1:

Fig 1b could benefit from a different visualization that allows better comparison of the measurements between samples. For example, a heatmap of samples x drugs could be helpful, perhaps just for those drugs measured in most (or all) samples, and for the top-hits identified in each sample, to see how they reproduce across samples.

Thank you for this comment and we agree if we had a complete drug panel for each patient sample that a heatmap would be a great way to show this data. However, some patient samples had limited cell number and we had to prioritize the number of drugs included in our CROCS (clinically relevant oncology combination screen) panel so the heatmap has lots of gaps. For this reason, we didn't include this in Figure 1 because we didn't like the visual impact of the gaps in the heatmap for Figure 1. We have included the heatmap of all the drugs for all the patient data in Fig. S3 (see below).

Figure S3: Heat map of CROCS HTDBP hits on primary MPM patient samples. Fresh primary MPM patient samples were dissociated, treated with CROCS and HTDBP carried out. Cells were analyzed by immunofluorescence microscopy and Z-score was calculated to identify drug/drug combinations that prime tumor cells. Heatmap show ranked (highest at the top) mean Z-score for each drug treatment (carried out in duplicate), for each primary MPM patient sample (MPS). Blue represents a hit with a Z-score ≥ 3 with no replicate <1.5 . Yellow are non-hits.

The authors should clarify - perhaps in the form of flow chart or table - which compounds were measured across which samples, as well as how and why certain compounds and combinations were prioritized for testing. This would make it easier to understand how the combinations for the in vivo follow-up were selected.

Agreed we needed to make this clearer. We have now included a Table (Table S3) which includes a list of all the compounds used in CROCS for each patient sample. As mentioned in the previous comment, some patient samples produced a limited number of cells, so we had to prioritize what drugs we wanted to test in those samples. It became clear that BH3 mimetic and/or PI3K/mTOR pathway inhibitors were top hits and of interest early on, so these compounds were ranked as higher priority for testing over compounds that didn't appear to prime MPM cells. Approximately 3 million cells are needed to complete the full panel of drugs in duplicate plus having enough cells for DMSO controls and BIM titration in triplicate. 7 out of 13 MPM patient samples had >3 million cells (Table S2). We were looking for common hits and this is how the top combinations were selected for in vivo testing (Table S7&8).

For example: Given that the samples on which only around 10 drugs were tested already contained the hits chosen to follow-up on, it is not entirely clear if the authors already knew the hit drug combinations they wanted to follow-up on starting their screen.

To clarify further, we had no idea what drug combinations top hits would be in the patient samples. However, when cell number was limited, we had to prioritize the drugs to be included in the screen, so we had to prioritize drugs that were top hits from the previous patient samples done previously. With each sample we gained more information of drugs that were common top hits and would adjust our drug prioritizes accordingly.

Another example: It appears the newer mTORC1/2 drug AZD2014 was only tested on later (higher letter) samples, but then still included in 'priority' panel of ~10 drugs tested on those samples with little material. Why? Did the authors already pick their hits at that stage and still screened more samples?

Originally AZD8055 was the only dual mTORC1/2 inhibitor used in the panel. When approximately half the samples had been profiled and the data was shared externally. An audience member mentioned that AstraZeneca were no longer perusing AZD8055, and the

clinically relevant mTORC1/2 inhibitor was AZD2014. At this point we thought it was important to add this to the panel right away since it was clear dual mTORC1/2 were important drugs that primed MPM in combination with BH3 mimetics. Hence why AZD2014 was only included in the samples with the higher letter (the higher the letter the later the sample was received and profiled).

Figures 1 and 3 suggest quite some patient sample variability. It would be good if the authors could analyze in more depth the clinical features associated with the patient samples which they screened, and their relationship to sensitivity to BCLi/mTORCi combinations. Were some samples perhaps obtained from patients that had received treatments already, whereas others may have been treatment-naïve? Are there pathology or genetic-level differences between these samples? This might explain in part observed patient variability that the plots suggest but not address, and greatly add to the novelty and interpretation of the presented work.

This is an excellent idea, but, in short, we simply did not have sufficient numbers in definable categories to make cogent comparisons. To help address this we included all the clinical data we have for each patient including histology, clinical stage at time of surgery, pre-surgery treatment, post-surgery treatment and clinical response of patient (Table S6). We also have the genetic oncopanel data for each patient but as mentioned in a previous comment we didn't find any aberrations that have clinical/potential clinical significance (Tier 1 and 2 variants) so cannot draw any correlations to drug sensitivities (Table S5). 4 out of 13 patients did have some prior therapy (before sample was taken) but 3 out of 4 of these samples produced limited number of cells (Table S2) so we had to prioritize the drug list was used (Table S3) and therefore we are unable to make any correlations between mitochondrial priming/drug vulnerabilities Vs. pre-surgery treatment or treatment naïve. As mentioned in a previous comment "Unfortunately, we only received one sarcomatoid patient sample and therefore, we cannot with see any differences between mitochondrial priming or top hits between epithelioid and sarcomatoid histologies. Even though biphasic tumors are made up of both epithelioid and sarcomatoid cells we wanted to see if there were any significant differences between epithelioid and biphasic patient samples. There were no significant differences between mitochondrial priming or top hits between epithelioid and biphasic histologies (Table S4)". All new supplementary Tables have been updated in the text.

For the combination screen drug treatment, the authors should clarify whether combinations were 1uM total concentration or 1 uM per drug. If it is 1uM for each drug, the authors should test if the effect they see for the combination is equal to the additive effect of each single drug.

We used 1 μ M for each drug in the combination. For single agents we used 1 μ M drug. We have updated the text “Cells were drugged using the HP D300e digital dispenser (Hewlett-Packard) at 1 μ M for each drug as a single agent or in the clinically relevant oncology combination screen (CROCS) “. We have looked to see if the hits are additive or synergistic and we have observed both additive and synergistic priming for same drug combination in different patient samples. We have not included this data in the manuscript but please see data below.

Fig 2:

Fig 2a and S3-5: Top hits consistently reduce the cell number - how does the BH3 priming z-score correlate with reduction in cell number across all screens? Is the BH3 readout necessary at all to identify these top hits?

There is certainly some correlation between treatments that reduce cell number and treatments that induce apoptotic priming. However, the correlation is not perfect (please see figure below). It is, however, difficult to answer experimentally the Reviewer’s question, although it is an important one. To do so would probably require an experimental design in which we performed in vivo testing of treatments ranked highest by a cell number metric and compared with in vivo

testing of treatments ranked highest by a delta priming metric. To make a statistically valid comparison would require a quantity of PDX studies that is beyond our resources.

We believe that the relative results, and the relative utility, of a delta priming metric and a cell number metric likely vary with context – both drug context and tumor context. However, we do have reasons behind our prioritizing delta priming over cell numbers. In our experience across many cancers, many drug treatments that cause apoptotic priming, which drugs we subsequently found to be active in vivo, do not cause cell number loss over the incubation period. For many drugs, it can take several days for measurable cell loss to occur, and we prioritize completing our ex vivo analysis in 24 hours to reduce factors of selection, adaptation, and overgrowth of non-malignant stromal elements. The main purpose of this paper was to demonstrate the ability of dynamic BH3 profiling to identify active combinations in this challenging solid tumor, but we cannot rule out the utility of alternative readouts.

Fig 2c and d: The visualization is difficult to read, maybe the plots could be replaced by dotplots / heatmaps to show the data per sample, and the mean z-score could be indicated as a barplot on top.

One of the main problems with using a heatmap for the patient data is that we had limited number of cells for 4 of the patient samples so we didn't run the full drug panel and the heatmap has many gaps in. However, we decided to include this in the supplementary (Fig. S3). We like the graphs in Figure 2C and D because it allows you to see inter-patient heterogeneity easily and you can follow the trends. For example, you can clearly see that single agents don't prime patient samples as much as the combinations do in Figure 2C.

Relating also to the flow questions on Figure 1: Panel d could be swapped with panel c, since I was initially quite confused how the drugs shown in panel c were chosen, and it made more sense after looking at panel d.

Thank you for this comment. As you suggested, we have changed the Figure 2c and d around and updated the text in manuscript to reflect this change.

Figure 2: BCL-xL and MCL-1 antagonism enhance priming with PI3K/AKT/mTOR pathway inhibitors in MPM. (a) Primary epithelioid MPM cells were treated as previously described in Fig. 1. Representative immunofluorescence microscopy images from MPM patient sample A (MPS:A). Images taken at 10-fold magnification. Hoechst 33342 used to stain DNA (blue) and identify the number of cells/well. Pan-cytokeratin-488 antibody (green) used to identify epithelioid cells (parent population). From the parent population, cytochrome c positive cells % was determined using cytochrome c-647 antibody (red). DMSO treatment is a negative control for cytochrome c loss. Non-hit is a drug treatment that didn't score above the hit threshold (no drug-induced priming). Top hit is the drug treatment that scored the highest mean Z-score (highest drug-induced priming) for this patient sample, first red dot in Fig. 1b. Scale bar is 100 μ m. (b) Schematic showing drug targets for BH3 mimetics, BH3 peptides and PI3K/AKT/mTOR pathway inhibitors used in this paper. All drugs (not BH3 peptides) shown here are included in CROCS list except A-1331852 (because it's not in clinical trials yet). BH3 peptides are peptides derived from the BH3 domain of the pro-apoptotic BH3-only Bcl-2 family members and are used in BH3 profiling/DBP assay. (c) Graph showing mean Z-score for top ten most common hits and hits with highest mean across all MPM patient samples. (d) Graph showing mean Z-score for navitoclax, S63845 or venetoclax in combination with PI3K/AKT/mTOR pathway inhibitors (combinations and single agents).

Fig 3:

Fig 3c: same comment as Fig. 1b, a drug response matrix/heatmap (even just of selected drugs) would greatly improve the readability of the per-sample drug responses the authors measure. Agree, changed Figure 3C to a heat map.

Fig 3d: same comment as Fig 2c/d. Moved Figure 3C to 3D.

Fig 4:

Fig 4f: The difference in the BIM peptide AUCs seem to come mostly from differences in baseline priming without BIM peptide added, as BIM EC50 is comparable across conditions. Can you then conclude that treated cells are more primed? Is baseline apoptosis the same as priming?

Thank you for this comment. While we excluded from analysis frankly dead, Hoechst-positive, cells at the stage of flow cytometry, there are still some cells that are in earlier stages of cell death that are not yet Hoechst positive, but have undergone mitochondrial permeabilization and cytochrome c loss. In the mitochondrial pathway of apoptosis, mitochondrial permeabilization precedes the loss of plasma membrane integrity detected by Hoechst. The cytochrome c loss is therefore an indirect indication of a cell population with high baseline apoptotic priming, cells that likely would have accomplished apoptosis in a few hours if left in vivo. These are cells that likely had their mitochondria intact when harvested, but during the assay lost mitochondrial integrity because they were so highly primed.

Is baseline apoptosis the same as priming? No, though populations that have a lot of baseline apoptosis often also have a lot of baseline priming. We consider a cell to be primed, but not apoptotic, if its mitochondria are still intact.

Note that the % of apoptosis in 24 hours for each treatment. BIM EC50 is significantly less in AZD8055 and navitoclax plus AZD8055 combination compared to the vehicle-control group (Table S11). Therefore, I can conclude that these treated cells are more primed than the vehicle-control. Manuscript edited to include the supplementary Table S11 with EC50 values.

Fig 5:

Fig 5 a-c: The number of replicates used for ANOVA is not specified

All in vitro work was carried out in triplicate. Figure legend updated to include this information

Fig 5 c: It is not clear which statistical comparisons are being made between the bars, as the

placement of stars is ambiguous.

Statistical comparisons are being made between the DMSO-control bar (red) and each of the treatment groups; this has been updated in the Figure legend. There are three sets of * for the three treatment groups (navitoclax, AZD8055 and combination. The * set furthest to the right is for the treatment group furthest from the right (purple/combination), second * set to the right is for the second treatment to the right (green/AZD8055) and third * set from the right is for the third treatment group from the right (blue/navitoclax).

Fig 5 d: The text specifies that AZD8055 increases BAK in 4/4 cell lines, and BAX in 3/4 cell lines, however the differences are in some cases small, and could be affected by differences in loading (e.g. with MSTO-211H). For some cell lines, the increase also appears to be reversed by co-treatment with Navitoclax. Similarly, it is unclear whether BIM increases with treatment in JMN1B cells as the bands are overexposed. Given the small effect size, I would therefore ask that the authors provide further replicates with a quantification of band intensity relative to loading control to allow for a more robust evaluation of these differences.

Thank you for this point and we agree. Now we have performed n=3 western blots (see below), and given the heterogeneity among the different cell lines, and the relatively small changes, we have eliminated mention of BAX and BAK, and have softened our language in describing the other trends, which are not completely consistent across all cell lines.

n1 24-hour drug treatment

n2 24-hour drug treatment

n3 24-hour drug treatment

Fig 5e: Could the differences in detected BIM be due instead to differences in total levels of MCL-1 and BCL-xL present (due to changes in expression or differences in loading)? For example, in figure S10, BIM levels appear to be elevated by Navitoclax in both the IP and S fraction. Further information on how the Western blots were loaded and performed should be provided in the materials and methods.

Yes, certainly BIM abundance does increase, which could be a challenge to interpretation. To address this, we performed quantitation of the BIM bands in the BCL-XL and MCL-1 IPs, and quantitated band intensity relative to the untreated controls in Figure 5E (H2052 cells) and Figure S13 (JMN1B cells). It can be seen that navitoclax or combination treatment increased the co-immunoprecipitation of BIM and MCL-1, while it decreased the co-immunoprecipitation of BIM and BCL-XL. See Figure below taken from Figure S13.

Figure S13: BIM complexes with BCL-xL or MCL-1 after treatment with navitoclax, AZD8055 or navitoclax plus AZD8055 combination in MPM cells. (a) 24 hours after treatment with indicated drug concentration, BCL-xL and MCL-1 were immunoprecipitated in JMN1B cells and BIM complexes were determined by Western blotting analysis (Input total cell lysate; IP, immunoprecipitated fraction; IgG1 (immunoglobulin isotype 1; MCL-1 isotype) and IgG3 (immunoglobulin isotype 3; BCL-xL isotype) control; S, supernatant). (b) Heat map of densitometry analysis carried out on BIM immunoblot for MCL-1 or BCL-xL IP for H2052 cells (Figure 5e) or JMN1B cells (part a of this Figure). BIM complexes to either MCL-1 or BCL-xL after 24 hours drug treatment with indicated drug concentration was calculated relative to DMSO-control. Relative values included in the heat map.

The reviewer also makes a good point, our diction regarding what immunoprecipitation indicates was sloppy, and we have changed the language to indicate that we are observing complexes rather than direct binding.

For IP we split the IP portion into 3 and load onto 3 gels to blot for BCL-xL, MCL-1 and BIM. On the same IP gel, we load 20 μ g of protein into the input and the supernatant wells. We have updated the methods to reflect this information. Figure S10 is now updated to Figure S13.

Fig 6:

AZD8055 likely has many more effects than downregulation of MCL1. To see if it really is MCL1 downregulation that causes the combined drug effect, it would good to do a knockdown of MCL1, and/or to perform transcriptomics on AZD8055 drug treated cells, to show that MCL1 downregulation is a key feature and most likely mechanistic explanation for the observed drug effect. If, in the end, the drug effect is merely additive, not synergistic, it is the interpretation on the mechanistic synergy proposed could be reduced as it is not critical for the study.

We didn't do MCL-1 KO on the MPM cell lines, but we did treat with a compound (triptolide) that downregulates MCL-1 protein levels to a higher degree than AZD8055 in vitro (Fig 5D Vs. Fig. S12A). Bcl-xL antagonism by navitoclax or A-13318452 further enhanced sensitivity to Triptolide compared to AZD8055 (Fig. S10C Vs. Fig S12D), showing the more MCL-1 is decreased the

more cells are sensitive to navitoclax. We showed above that drug-induced priming with navitoclax plus AZD8055 can be both additive or synergistic in different patient samples.

Materials and methods:

The “Tumour cell lysis and Immunoprecipitation Assay” requires more detail on how the western blot was conducted. Was the entire IP and supernatant loaded in each condition, or were samples normalised to total protein content etc.?

Updated this section in the test to include the amount of protein loaded for the supernatant and input and how the IP sample was divided into 3 for 3 gels to blot for MCL-1, BCL-xL and BIM.

Minor comments

Writing comments / typos etc:

The manuscript feels a bit rough, including typos, and small mistakes in figures and citations. Below is a non-exhaustive list, it would be great if the authors could go over their manuscript carefully to double-check and correct such issues.

(L28) “in a MPM” -> “in an MPM” updated in text

(L77) “anti-apoptotic’s” -> “anti-apoptotics” updated in text

(L107) Please keep the interpretation on “increased ATP production”, not “increased viability”. updated in text

(L108) “RMPI” → “RPMI” updated in text

(L164) “to devoid” → “to be devoid” updated in text

(L188+) “mg/kg/qd” -> “mg/kg/d” or “mg/kg qd” updated in text

(L503) “standard deviated” -> “standard deviation”

Axis label in Fig 4b: “Tumor volume > 5xITV” -> “Tumor volume < 5xITV” I want it to be more than or equal to 5xITV so changed to tumor volume \$\geq\$ 5xITV

References: Reference 21 and 23 are the same paper. updated in text

Please rephrase Figure S1 legend: “A BIM dose response on untreated primary MPM tumor cells was carried 16 hours after seeding, to calculate the optimum BIM BH3 peptide concentration. The point where MOMP is about to occur (BIM EC10) and drug treatments that prime MPM cells would readily release cytochrome c (surrogate for priming), after being incubated with optimum BIM BH3 peptide concentration.”

Please rephrase Figure S3-S5 legend: “Hoechst 33342 used to stain DNA (blue) and therefore identify the number of cells present in each well. Pan-cytokeratin-488/vimentin-488 antibody

(green) used to identify sarcomatoid cells (parent population).”

Thanks for this, very helpful. We have made the above changes in the text and re-proofread the manuscript.

Reviewer #3 (Remarks to the Author):

In this interesting and thought provoking manuscript the authors examine the utility of high throughput dynamic BH3 profiling for identifying drug combinations that effectively kill mesothelioma in primary cultures and PDX models. The authors have pioneered this technique and should be lauded for moving it towards clinical utility.

The results of the study are very interesting and it appears that the combination of AZD8055 and Navitoclax not only confers ‘priming’ but that this priming translates to tumor kill in PDX models. Overall I am very enthusiastic about most of the results in the paper. My main problem is with the molecular model proposed to explain the increased efficacy of the combination.

One of the major conclusions from the paper is: “Mechanistic investigation revealed that AZD8055 treatment down regulates MCL-1 protein levels, increases BIM protein levels, and increases MPM mitochondrial dependence on BCL-xL, which is inhibited by navitoclax. Navitoclax treatment increases dependency to MCL-1 and increases BIM protein levels.” These statements are based to a large degree on measurements of BCL-XL, MCL-1 and BIM protein levels by western blotting (data shown in Figure 5D-E and supplementary Figure 10). To my eye in Figure 5D addition of Navitoclax not AZD8055 increased BIM protein in H2052 cells, both Navitoclax and AZD8055 increased BIM in JMN cells, JMN1B the blot is overexposed and therefore uninterpretable, and Navitoclax increased BIM more than AZD8055 increased BIM in MSTO-211H cells. MCL-1 levels were unaffected in H2052 and decreased to varying degrees by AZD8055 in the other cell lines. The patterns are also inconsistent with Figure 4e (see below). The magnitude of changes looks similar for BAK and BAX but those changes are also not correlated across cell lines. The most striking observation is the decrease in pAKT as a result of AZD8055 addition.

In 5e the immunoprecipitates are also difficult to interpret. It appears that Navitoclax increases protein levels for MCL-1 and BCL-XL and that results in more co-precipitation of BIM with MCL-1. This is not a surprising result and I am pretty sure that BIM binding to MCL-1 has been reported to stabilize the proteins. I remind the authors that co-precipitation does not indicate direct binding despite what is claimed in the manuscript.

Thank you for this point. Reviewer 2 made the same comments, please see our response above.

In addition, the cell lysis buffer used contains Triton, a detergent known to artificially promote interactions between Bcl-2 type proteins as elegantly shown by Youle years ago.

Good point. We do not use any Triton based lysis buffer for IP lysis buffer. We use 1 % CHAPS lysis buffer for IP to prevent artifactual activation of BAX and subsequent promotion of artificial interactions, as shown by Youle many years ago. We have updated this in the methods and materials section to clarify this.

Figure S10 shows no Bim binding to Bcl-XL (the size change suggests the band shown is a non-specific band) and even when MCL-1 decreased a lot, it bound the same amount of BIM when cells were treated with the mTor inhibitor as without the mTor inhibitor.

However, since none of this was quantitated, no replicates are provided and only thin slices of blots are presented, interpretation of the western blots is really difficult.

We agree that there is some heterogeneity in the magnitude of effects of MCL-1 alterations in abundance, BIM levels, and other BCL-2 family proteins in this study, yet each cell line tested contains some of these elements. While there is doubtless variability, our interpretation of the sum of the data is that navitoclax increases priming and dependency to MCL-1 as BIM complexed with Bcl-xL decreases, BIM complexed with MCL-1 increases. We confirm this with our BH3 profiling data showing the increased response to MS1 peptide (Fig. 5A), indicating increased dependency to MCL-1 after navitoclax treatment. Building on this, AZD8055 increases response to BAD peptide (navitoclax is a BAD mimetic; Fig 5B).

Fig. S10 is now Fig S13. We don't think the BIM band in Bcl-xL IP is non-specific because we do not see it in the IgG control for the Bcl-xL antibody (IgG3 isotype). While we cannot explain the specific biochemical alteration responsible, you can see a similar migration change in some of the lanes of the western blots of BIM in H2052 in the Western blots provided above. BIM levels increase in the MCL-1 fraction, after navitoclax-only or combination treatment for both H2052 and JMN1B, even though MCL-1 levels go down in JMN1B cells for the combination input fraction and IP fraction and slightly up for H2052, suggesting this is due to an increase in

MCL-1 complexed to BIM and not just changes in MCL-1 levels. Please see quantitation of IP's above in response to Reviewer 2 comment taken from Figure S13B. Full IP blots are below.

H2052 IP full blots for Figure 5E (Input and IP fractions)

H2052 IP full blots for Figure 5E (Input and Supernatant fractions)

JMN1B IP full blots for Figure S13 (Input and IP fractions)

JMN1B IP full blots for Figure S13 (Input and Supernatant fractions)

Minor points –

Are all the abbreviations really necessary? The result is phrases like “followed by HTDBP on CROCS-treated primary MPM cells” which in the absence of a glossary are very difficult to follow.

Thank you for this comment and we agree we have many abbreviations, but we feel they are necessary. The first time we mention or start a sentence with the abbreviation we type out the abbreviation. We say high throughput dynamic BH3 profiling (HTDBP) 27 times, clinically relevant oncology combination screen (CROCS) 17 times and malignant pleural mesothelioma (MPM) 109 times throughout the manuscript. To spell these abbreviations out every time, will increase the word count by 547. We will happily include a glossary if it is permitted by the editor.

Another highlighted conclusion is that “These findings validate the principle that HTDBP can be used as a functional precision medicine tool to rationally construct combination drug regimens in MPM and other cancers.” However, it is not clear how much primary cancer tissue is required. Based on the amount of PDX material required (a tumor diameter of about 1 cm) it would seem that these analyses are not practical for tumors from which the tissue sample would be limited to one or two fine needle biopsy samples. Instead, the analyses are limited to tumors for which tissue is obtained during resection. Since the authors propose that the current experiments with mesothelioma are exemplary for other tumor types the limitations imposed by the tissue requirements should be discussed by the authors.

Great point. We show the cellularity and size of the patient samples used in the study in Table S2. The smallest sample was 355 mg. Ideally to complete a full panel of drugs in the screen ~3 million cells are needed but if you limit the number of drugs/prioritize drugs you can profile with much fewer cells. The lowest number of cells retrieved from a sample was 170k cells which is comparable to the number of cells our lab has seen with some needle biopsies. If a core biopsy produced enough tumor cells, then HTDBP could be carried out with a prioritized, limited number of drug combinations. Also, we did run ex vivo HTDP on two MPM PDX tumors that were ~500 mm³ (Fig. 3a) and this produced enough cells for the full CROCS drug panel.

However, we acknowledge that reducing the number of cells required is desirable, and in fact, we have been working hard at this in the laboratory. In a separate study, we have shown that we can supply very similar information using dramatically fewer cells (e.g., roughly 10-fold

fewer) using a strategy we call Ramp Up DBP. We are preparing a separate publication on that now. In the meantime, we have included the following in the Discussion:

“These studies relied on tissue from resections of MPM, as resections are common practice in this disease. In other tumor types, resection tissue may be more limited, and provided mainly by needle biopsies. We hope to report soon our work on performing experiments like those in this paper using far fewer input cells via a modification of the technique employed here.”

Figure 1c shows the correlation of the drug responses for two samples from different tumors in the same individual. While the correlation is obvious, it is not clear how correlated the results are between patients for examples between MPS:C and MPS:H or MPS:H with MPS:L. Thank you for this comment. We didn't look at direct correlation between MPM tumor samples from different patients because we are most interested in identifying common drug combination hits across all patient samples. The main point in Fig. 1D is to compare MPS:L and MPS:M is to see if chemical vulnerabilities correlated, given the opportunity of two tumor samples from one patient.

In Figure 4e adding AZD8055 doesn't increase CI PARP or decrease pS6 or increase CC3 there is just a small increase in Mcl-1 with Navitoclax and a small decrease in MCL-1 without Navitoclax and a decrease in pS6. The authors should comment more on the significance of this data.

AZD8055 addition does appear to reduce pS6 levels compared to vehicle treated or navitoclax only. Regarding preservation of MCL-1 in the combination, it is possible that cells that remain still have MCL-1 and that there has been some selection for survival of such cells. We have added some sentences in the text to explain. See edited text below.

“To investigate the in vivo mechanism of action of the navitoclax plus AZD8055 combination in MPM, CPDM_0011x PDX tumor bearing mice were treated with one dose of either: 1) vehicle, 2) navitoclax-only, 3) AZD8055-only or 4) navitoclax plus AZD8055. Twenty-four hours later tumors were harvested, dissociated, and used for Western blotting analysis. PARP cleavage and caspase 3 cleavage were observed, consistent with an in vivo apoptotic mechanism of tumor cell death. MCL-1 is a known resistance biomarker for navitoclax treatment and AZD8055 has been previously shown to reduce MCL-1 protein levels. AZD8055-only decreased MCL-1 protein levels

compared to untreated in the CPDM_0011x PDX cells but had no effect on the apoptotic biomarkers cleaved PARP and cleaved caspase 3 suggesting a reduction in MCL-1 levels alone is not sufficient to induce apoptosis in this model (**Fig. 4E**). Note that the navitoclax plus AZD8055 combination arm demonstrated preserved MCL-1 levels at the 24-hour time point. This is possibly because tumor cells that most downregulated MCL-1 via inhibition of mTOR pathway (AZD8055) in combination with navitoclax have undergone apoptosis (high levels of cleaved-PARP and cleaved-caspase 3 **Fig. 4E**) resulting in removal of dead cells from the tumor by phagocytosis. Phospho-S6 is used as a pharmacodynamic biomarker for AZD8055 activity and levels are reduced in arms treated with AZD8055 (**Fig. 4E**).”

In Figure 5a-b In general there isn't a big change with Bim peptide plus AZD8055 across cell lines. The more impressive changes are for PUMA and BAD. Is that due to the differences in peptide concentrations or are the large differences in peptide concentrations reflective of variations in peptide binding? What do the authors make of these differences?

We think much of what the Reviewer observes is simply because many of the BIM concentrations used are outside of an informative range. For instance, BIM at 3 μM by itself causes very high cytochrome c release even in the absence of AZD8055 treatment, so any change caused by drug treatment is hard to observe using BIM at 3 μM . Changes can be observed with the BIM peptide when we use it in a more informative range, such as 0.3 μM .

In Figure S9 the decrease in luminescence due to the addition of Navitoclax or A-1331852 to AZD8055 is not very impressive. How does this fit with the model being presented?

With the addition of new Figures, old Figure S9 is now Figure S10. Agreed, not the most impressive with this assay at this time point (24 hours). Sometimes, especially if a perturbation can potentially influence energy metabolism, Cell Titer Glo can be an imperfect measure of cell viability. However, we still see a down shift in drug response indicating some increased sensitization to AZD8055 when Bcl-xL is antagonized in the combination (navitoclax or A-1331852) compared to AZD8055 alone. We start to see a significant difference when we treat with a lower concentration of AZD8055 for longer (72 hours) with navitoclax, as shown with the colony formation assay in Figure 5C.

Figure 5C

Reviewer #4 (Remarks to the Author):

In this paper, the authors apply their “dynamic BH3 profiling” to identify combination therapies for mesothelioma. This method allows them to assess apoptotic priming as a correlate of efficacy, permitting the use of primary patient samples rather than culture adapted cell lines. Here they show the utility of the approach and provide compelling data that the results are highly reproducible across multiple tumors, that primary tumors and PDX tumors behave similarly, and that the results have application to in vivo therapy in preclinical models. The paper is a tour de force of the use of this methodology.

Before I go into details, I feel it is important to state that while the top hit they produced is not especially surprising (it has been known for some time that mTOR inhibition reduces Mcl-1 levels and sensitizes for killing by Navitoclax), the fact that this emerged from dynamic BH3 profiling of primary specimens is exciting and important. It also points to a significant finding that I discuss in more detail below.

My relatively minor questions and concerns follow in no particular order.

1. As mentioned above, the fact that cross-specimen comparisons show high correlations is striking (and exciting), as is the correlation with PDX. While the combinations that represent “top hits” would also be expected to appear in cell lines analyzed in a similar manner, it would be helpful to know the extent to which the same panel of agents and combinations show less correlation between primary samples and cell lines (such as those they used). If so, it would add impetus for the use of primary samples and dynamic BH3 profiling for drug combination, therapeutic discovery.

Thank you for your comment. We have included data file S3 which includes all the hits that correlate between MPM patient samples and PDX tumors, as well as hits found in the patient and not the PDX tumors, and hits found in the PDX but not the patient samples. Most of the hits that are found in patient samples and not in the PDX tumors, were not common hits across most patient samples (more than 50% being common) except for two drug treatments (1- crizotinib plus AZD8055; 2- navitoclax plus paclitaxel). We decided to focus on common hits across patient and PDX's because we wanted to prove proof of principle that HTDBP can identify hits ex vivo and these hits are efficacious in vivo in MPM PDX. Patient cells are the gold standard model to use, and we are very lucky to have access to such precious samples. We did not employ established cell lines for CROCS HTDBP.

2. The finding that Bim protein levels are elevated by the treatment is important (and is likely to have mechanistic value). Since Bim is regulated at both transcriptional and translational (stability) levels, it would be helpful to know if the upregulation is at the mRNA level (or not). Great point. To address this, we carried out qPCR after treatment with AZD8055, navitoclax or the combination at 24 hours on a panel of BCL-2 family members (Figure S11B and shown below). We saw no increase in BIM mRNA levels suggesting it was translational or post-translational regulation of BIM responsible for protein levels increasing after treatment. We agree this increase in BIM has mechanistic value as shown in Figure 6 schematic.

Figure S11: Quantification of protein and mRNA levels for BCL-2 family members after treatment with navitoclax, AZD8055 or navitoclax plus AZD8055 combination in MPM cell lines. MPM cell lines (H2052, JMN, JMN1B and MSTO-211H) treated with the indicated concentration of navitoclax, AZD8055 or navitoclax plus AZD8055 combination for 24 hours. Graph showing quantification of $n=3$ proteins levels by Immunoblots (a) or $n=3$ mRNA levels by qPCR (b) in MPM cell line after 24 hours treatment with indicated drug concentrations for BAK, BAX, BCL-2, MCL-1, BCL-xL, BIM and PUMA. BCL-2 family members were normalized to loading control (immunoblots) or house keeping gene (qPCR) and then shown relative to DMSO-control levels for that specific protein (a) or gene (b). Significance was calculated using a 2-way ANOVA multiple comparisons test to DMSO-control, * $p < 0.05$, ** $p < 0.01$, *** $p < 0.001$ and **** $p < 0.0001$.

3. Perhaps the most important finding is that while the combination of Navitoclax with the Mcl-1 inhibitor S6 is highly lethal in vivo, the combination of the mTor inhibitor with Navitoclax is well tolerated. (The rapidity of death-4 hrs-with the Nav/S6 combination suggests possible cardiotoxicity, which has been described). Of course, Torc1/2 inhibition and BH3-mimetic treatment have different mechanisms of action on Mcl-1; the first relies on the short half-life of Mcl-1. This leads to a fundamental question: does the mTOR inhibitor sensitize for Navitoclax without depleting all Mcl-1 (i.e., is there a stable pool of the protein that protects the animal but not the tumor?). Indeed, the disparate results on Mcl-1 levels in Fig. 5e (not reduced with AZD8055) and Fig. S10 (reduced with AZD8055) are suggestive and should be discussed. If possible, it might be useful to assess levels of Mcl-1 expression and extent of cell death in hearts of animals treated with the two combinations (since we do not know the mechanism of toxicity, this may be asking too much, but it may be worth a look).

Thank you for this point and great question. To your point of the result between Fig. 5E and Fig S10 (now Fig. S13). We included a cell line that AZD8055 didn't reduce MCL-1 levels (H2052, Fig 5E) and a cell line where it significantly did reduce MCL-1 levels (JMN1B, Fig. S13) after AZD8055 single agent and combination treatment. In both cases where MCL-1 levels were either significantly reduced (Fig. S13) or not (Fig. 5E) in the combination group we observed an increase in BIM found in the MCL-1 IP fraction but not in the AZD8055 alone treatment suggesting an increase in BIM bound to MCL-1 (irrespective of changes in MCL-1 protein levels).

Unfortunately, because the combination was so unexpectedly toxic for navitoclax plus S63845, in such a short time, the vet removed the mice after they had died due to severe toxic events. This meant we were not able to carry out a necropsy and assess any of the mouse tissue. Knowing how toxic the combination now is, it is unethical and outside the scope of our animal protocol to knowingly dose mice with such a severely toxic drug regime to gain tissue to assess MCL-1 expression in various mouse tissue parts. We agree this would be interesting to know, but out of the scope of this study. Understanding in vivo toxicity of MCL-1 inhibitors is under current investigation in several significant industrial studies.

Reviewers' Comments:

Reviewer #1:

Remarks to the Author:

This novel study shows that targeting both BCL-XL and MCL-1 in mesothelioma can engage mitochondrial apoptosis. This work has therapeutic potential, and is therefore of relevance for the field of mesothelioma research, a cancer of unmet need.

The findings are significant, have relevance in the field and could be translated clinically.

The work is original and builds on a well defined research methodology (BH3 priming) which was pioneered and has been established over several years by the lead author. It is of considerable interest that it has been applied in the setting of mesothelioma, yielded results which could be potentially taken forward in the clinical setting.

The work supports the conclusions and claims that have been made. I am satisfied that the authors have provided a number of updates to the original manuscript, which address all of my initial criticisms and provides a significant improvement on the previous version.

The work is generally of high quality and is unique in the field of mesothelioma research. The quality of research is sound, with the refinements to this version of the manuscript having greatly improved the paper's quality.

I feel the detail provided in the methods are sufficient to allow for the experiments in this work to be reproduced

Reviewer #2:

Remarks to the Author:

The authors have fully addressed my remaining concerns.

Reviewer #3:

Remarks to the Author:

The authors have revised the paper satisfactorily and it is now ready for publication.

Reviewer #4:

Remarks to the Author:

The authors have fully addressed my comments.